# A universal system for boosting gene expression in eukaryotic cell-lines

Inbal Vaknin[1], Or Willinger[1], Jonathan Mandl[2], Hadar Heuberger[3], Dan Ben-Ami[3], Yi Zeng[1], Sarah Goldberg[1], Yaron Orenstein[2,4] & Roee Amit [1,5] ✉

We demonstrate a transcriptional regulatory design algorithm that can boost expression in yeast and mammalian cell lines. The system consists of a simplified transcriptional architecture composed of a minimal core promoter and a synthetic upstream regulatory region (sURS) composed of up to three motifs selected from a list of 41 motifs conserved in the eukaryotic lineage. The sURS system was first characterized using an oligo-library containing 189,990 variants. We validate the resultant expression model using a set of 43 unseen sURS designs. The validation sURS experiments indicate that a generic set of grammar rules for boosting and attenuation may exist in yeast cells. Finally, we demonstrate that this generic set of grammar rules functions similarly in mammalian CHO-K1 and HeLa cells. Consequently, our work provides a design algorithm for boosting the expression of promoters used for expressing industrially relevant proteins in yeast and mammalian cell lines.

Production of proteins at scale and affordable cost has been a major need of the biotech sector for the last several decades. This need was one of the primary drivers that led to the development in the 1970's of recombinant DNA technology, which in turn revolutionized the pharmaceutical sector by facilitating the creation of biologics or protein-based therapeutics, the first being human insulin expressed in *E. coli* and industrially produced by Genentech. In recent years, a new sector has emerged with major needs in manufacturing of affordable proteins at scale primarily for the food industry. This sector, referred to colloquially as "alternative proteins" or "precision fermentation", aims to replace most traditional protein sources for food products with more efficient and climate-resistant sources of food-based proteins. While some of these alternative proteins are expected to be sourced from widely available plants, others are expected to be manufactured via fermentation of engineered microbial species, such as the traditional biotechnology work-horse *E. coli* bacteria, and various yeast species including the baker's yeast *S. cerevisiae* or the more industrially relevant *Pichia pastoris*.

Historically, over-expression of proteins leading to larger titers of relevant protein mass at the fermentation scale had only been achieved in bacteria via the T7-promoter system[1]. This expression system is based on a monomeric phage-sourced RNA polymerase, whose promoter is capable of initiating transcription of mRNA at a much higher rate as compared with the endogenous bacterial RNA polymerase. This, in turn, leads to a huge boost in steady state mRNA levels and results in high protein titers per cell. The T7-promoter expression system was further developed over the years to yield even higher expression levels[2], potentially optimizing the system's output. Unfortunately, many industrially relevant proteins cannot be expressed in bacterial cells. This is because eukaryotic proteins frequently need to undergo a myriad of post-translational modifications that can only occur in eukaryotic cells (e.g., glycosylation), which play a critical role in determining the protein's function. To date, neither T7 nor a similar high-level protein expression system has been shown to work in eukaryotic cell lines. While design of such systems in yeasts[3,4] and fungi[5] were attempted, they did not yield measurable protein titers and as a result generated limited progress on this complex problem. In addition, precision fermentation in most non-bacterial organisms is limited by a lack of advanced gene expression tools that can facilitate over-expression or a wide range of expression levels of genes in the relevant host cells. Therefore, a major challenge for the field is to discover and/or develop genomic tools that will enable

[1]Department of Biotechnology and Food Engineering, Technion, Haifa, Israel. [2]Department of Computer Science, Bar-Ilan University, Ramat Gan, Israel. [3]School of Electrical and Computer Engineering, Ben-Gurion University of the Negev, Beer Sheva, Israel. [4]The Mina and Everard Goodman Faculty of Life Sciences, Bar-Ilan University, Ramat Gan, Israel. [5]The Russell Berrie Nanotechnology Institute, Technion, Haifa, Israel. ✉e-mail: roeeamit@technion.ac.il

over-expression of proteins in various eukaryotic cell lines, thus yielding larger protein-titers at the fermentation scale.

In eukaryotic cell-lines, gene expression levels are regulated by transcription factors (TFs) that bind to DNA regulatory motifs, together with chemical, structural, and spatial changes in the chromatin state, and binding of the transcriptional machinery to the core promoter. DNA regulatory motifs serve as cis-acting TF binding sites (TFBSs) for activators and/or repressors. The combination of different repressing and activating DNA regulatory motifs enables tuning promoter activity and, as a result, gene expression level. The sequence space of combined DNA regulatory motifs is often referred to in the literature as the regulatory code or grammar[6–8].

Over the past two decades, many systems and synthetic-biology studies attempted to decipher the regulatory code with varying degrees of success using two approaches. In the first approach, large regulatory regions were dissected using a traditional knock-down and rescue approach until the regulatory effect of every active TFBS was characterized[9–11]. In the second approach, a multitude of synthetic cis-regulatory regions (e.g., synthetic enhancers) composed of a small number of TFBSs arranged in various configurations were encoded in an oligo library (OL) and characterized using massively parallel reporter assays (MPRA) such as SORT-seq[8,12,13]. These studies found that it was possible to increase the expression of a particular target gene by positioning a cassette of repeat TFBS immediately upstream of minimal core promoters. This increase or "boost" in expression was reported in multiple studies[12,14–16], and is used in artificial gene-expression systems such as the ubiquitous inducible mammalian rTet-On system[17]. However, the current expression "boosting" systems are far from optimal. They are often utilized with a limited number of core promoters (e.g., rTet-VP16 together with the minimal CMV promoter), require the creation of a stable cell line expressing the synthetic TF, and frequently poorly translate to other cell lines. Other constitutive strong promoters, such as the industrially relevant EF1a and PGK promoters, become fully activated based on the epigenetic state or carbon state of the cells, respectively. However, growth conditions that are needed to trigger full activation may not be compatible with industrial requirements during fermentation[18,19].

Given the advances in the understanding of the regulatory grammar over the past two decades, we hypothesized that it was possible to develop a widely applicable or universal gene-expression boosting system. To be designated "universal", the system must function similarly in diverse eukaryotic cell lines, at several growth conditions, and with multiple core promoters. The goal of such a universal boosting system will be to substantially increase expression of proteins over the baseline core promoter's expression level and independent of the cell line used for the industrially relevant application, thus allowing the user to increase the protein titers in the final fermentation-scale process.

To develop a generic boosting design algorithm, we designed a simple regulatory-code architecture composed of a non-descript synthetic minimal yeast promoter and a synthetic upstream regulatory region (sURS). We populated the sURS with up to three TFBS motifs (from a list of 41) that are, for the most part, conserved across the eukaryotic lineage, thus allowing us to encode a large OL for the purpose of characterizing the boost potential of each motif either by itself or in combination with other motifs. Using bioinformatic and machine-learning (ML) analyses, we were able to classify active motif function into either "boosting", "attenuating", or "undetermined" for each of the 41 motifs. The characterized motifs allowed us to construct a computational model, which enables prediction of expression level boosts. We tested and validated the model on several tens of previously unseen sURSs in yeast and mammalian cells. Consequently, our motif-encoded sURS library yielded a regulatory architecture and associated design algorithm for non-inducible boosting of gene expression in yeast, mammalian cells, and potentially many more cell lines.

## Results

### Motif-based design of a synthetic URS oligo library

We constructed a motif-based sURS OL for yeast cells by encoding 41 DNA regulatory motifs mined from the following organisms: *S. cerevisiae*, *S. pombe*, *D. melanogaster* S2 cells, and mouse (ES cells and different tissues) (Fig. 1a, Supplementary Data 1, Supplementary Fig. 1). We reasoned that conserved motifs are more likely to function similarly in most organisms, and can thus form the regulatory backbone for a universal boosting system. To increase the likelihood for successfully identifying universal "boosting" motifs, we encoded the sURS library using a mixed-base OL synthesis approach[20], in which specific mixtures of 2 nucleotides are added in the same synthesis step at selected positions. The mixed-base motifs are composed of two to sixteen sub-motifs (1 to 4 composite bases), which constitute a first low-complexity attempt at studying position-weighted matrices (PWMs) in a synthetic regulatory context. Specifically, 17 of the motifs were previously classified as repressing or activating motifs in their relevant source cells, and 10 motifs were previously found to have dual-regulatory functionality (i.e., they can act as either repressors or activators, depending on the promoter and the other TFs with which they interact). Finally, 14 motifs were recently discovered in a large binding screen[21], and their regulatory effect in yeast is unknown (see "Methods" for details about rationale of motif choice).

To encode a first approximation to the PWMs of our selected motifs[21], we incorporated K (G/T) and M (A/C) mixed bases in 20 of the motifs, at positions where the two nucleotides represented by either K or M were likely to appear (>73%, see "Methods"), while using the most probable base for the remaining positions (Fig. 1b). Overall, 189,990 barcoded oligos were designed according to the following characteristics: multiple barcodes for each variant, number of motifs in the variants (0, 1, 2, or 3 motifs), motif's K/M substitutions, and the order of motifs in each variant. For 2435 of the oligos, picked randomly, we generated 22 barcodes, while for the rest of the oligos we generated 2 barcodes. To ensure the same background expression level for all variants, we designed a "desert" chassis in silico to exclude any known yeast *S. cerevisiae* TFBS motifs. To generate this desert sequence, we computationally excluded all known consensus yeast motifs as specified in the YeTFaSCo database[22]. In all the variants, the motifs were embedded within the desert sequence, at fixed positions, with 17 bp spacing between motifs. Each variant in the OL consists of a variable sURS, which regulates the mCore1 minimal promoter[23] activity driving yeCitrine expression (Fig. 1c). The OL (Twist Bioscience) was amplified using PCR, then cloned into a plasmid and amplified in E. cloni 10G electrocompetent cells (see "Methods" for more details). Mid-cloning deep sequencing was performed to evaluate the OL coverage (Supplementary Fig. 2). Purified plasmids were linearized and integrated into the yeast URA3 genomic locus (Fig. 1d). After integrations, the cells were grown in selective media for 3 days and sorted into 4 bins by fluorescence-activated cell sorting (FACS), according to yeCitrine fluorescence (see "Methods" for details). Finally, genomic DNA was extracted from each bin, the OL region was amplified and bin-barcoded by PCR, and sent to next-generation sequencing (NGS) (Illumina Nova-seq) (Supplementary Fig. 2). Overall, we identified ~400 M valid reads, which allowed us to retrieve 147,731 variants (of 189,990 designed) from the OL (Supplementary Data 2).

Upon initial analysis of the NGS data, we discovered three separate groups of retrieved variants, distributed across a wide range of mean fluorescent expression level (Expression (A.U.) - (-500–7500)). This indicates that the OL encodes a broad range of regulatory activity of yeCitrine expression (Fig. 1e). The groups were defined as follows: Group 1 contains 107,868 variants with a wide range of mean fluorescent expression level (as shown in Fig. 1e) with at least 40 total reads (i.e., variants with total number of reads greater than the minimum observed in Fig. 1f). Group 2 corresponds to the 28,431 variants whose reads are above 40 and appear predominantly (i.e., >90%) in the upper bin, and thus correspond to "saturated" or under-sampled mean

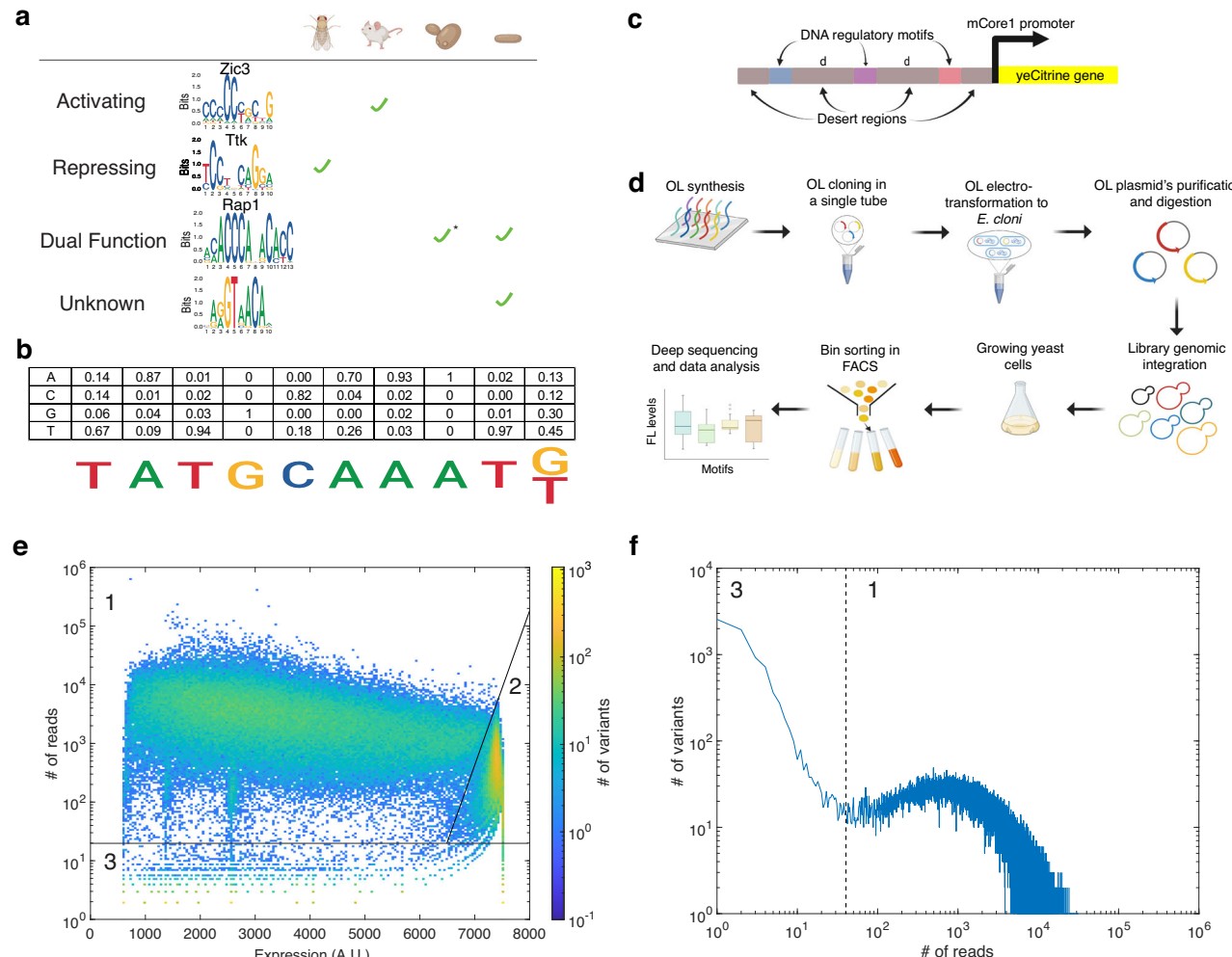

**Fig. 1 | Motif-based sURS OL-schematic and expression datasets. a** Sequence logos of motifs used in the sURS OL, their function (if known), and source organism. The organism corresponding to the sequence logo is shown in the figure. **b** Conversion of a PWM to the mixed-base motif used in the OL, with the Pou51 motif from mouse ES cells in the figure. (**c**) Promoter design consisting of a synthetic URS encoding three motifs within a desert sequence devoid of binding sites. The sURS is positioned upstream from the mCore1 promoter that can drive expression of yeCitrine at a low level. **d** Schematic for OL assembly and experimental

characterization. For more details, see review[40]. **e** 3D-heatmap distribution for the number of variants obtained for an integer number of reads with a particular mean fluorescent expression level. Of the 189,990 variants encoded, 147,731 were detected and analyzed. **f** A 2D-frequency distribution showing the number of variants obtained for a particular number of reads. In our analysis, we used only variants with a high number of reads (i.e., >40), which are located in region 1. Created with BioRender.com. Source data are provided as a Source data file.

fluorescent expression level range. Group 3 corresponds to all variants with <40 reads, and contains 13,108 variants in total. Group 1 demonstrates a Gaussian-like read distribution consistent with a well-sampled library, while Group 3 has an inverse exponential behavior resulting from a group of under-sampled variants. Group 1 also contains single-bin variants (bins 1–3), apparent as protrusions at mean fluorescent expression level values of 600, 1300, and 2600, likely indicating fluorescence from a small number of cells (despite having >40 reads). We proceeded with only Group 1 variants in subsequent analysis.

## Variants manifest a broad range of regulatory behavior

We analyzed the unsaturated variants with high read count (Group 1, 107,868 variants in total). First, for each motif (41 TFBS motifs and the desert motif) we grouped all variants containing that motif at least once and calculated the mean fluorescent expression level distribution for each group. The results are shown as individual boxplots in Fig. 2a, arranged by decreasing order of the median computed for each distribution (red lines), where the green line delineates the median computed for the desert motif's mean fluorescent expression level distribution. Figure 2a shows a dependence of the median values on

the motifs, which vary from <2000 for the lowest median to ~3000 for the highest median. When comparing the group of variants containing each motif to the group of variants containing motifs that are ranked by the median at least 5 motifs away, we observed significant differences for all motif-containing groups except for motifs 16-21 (out of 42) in decreasing order ($p$-value < 0.05, two-sided Wilcoxon rank-sum test, Benjamini-Hochberg-corrected with FDR = 0.1). Interestingly, the median of the desert motif distribution scored at the higher end of the median scores, indicating that the majority of motif distributions generate a reduced median expression score with respect to the desert motif reference. The desert variant itself displays lower expression level, as expected (black line in Fig. 2a). Therefore, motif mean fluorescent expression level distributions positioned at the top and bottom of the list are more likely to be enriched with up-regulating and down-regulating binding sites, respectively.

We next calculated the Pearson correlation between the medians of the mean fluorescent expression level distributions and the number of variants in each distribution (Fig. 2b). We found a positive correlation of ~0.3 between the median values and the number of variants within a certain motif distribution. This correlation suggests that motif

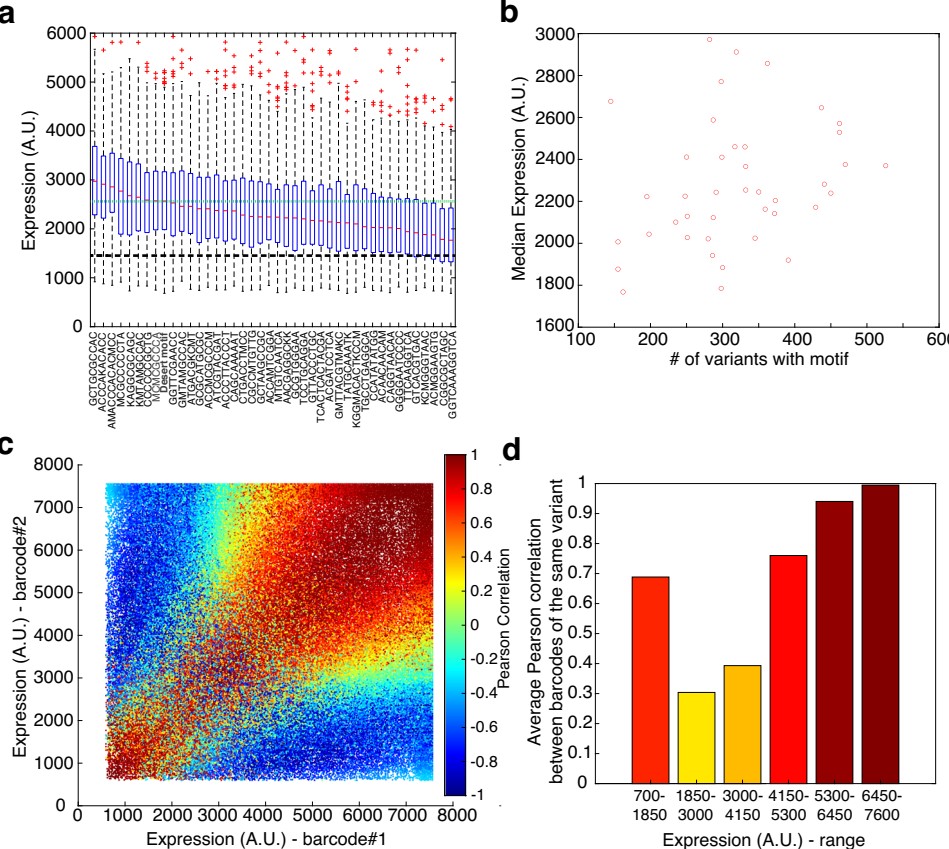

**Fig. 2 | Empirical analysis suggests that there are functional up- and down-regulating motifs. a** Box-plot distributions of mean fluorescent expression level of all variants containing at least one occurrence of a specific motif. Boxplots indicate a statistically significant difference between the motifs positioned at the left and the right of the plot, supporting a regulatory function. Black dashed line represents the mean fluorescent expression level of the 2 desert variants in the OL, while the green dashed line represents the mean fluorescent expression level of the desert motif. Desert-motif variants contain 2 out of the 41 motifs with a desert region in the middle of the variant as a third "motif". On each box, the central mark indicates the median, and the bottom and top edges of the box indicate the 25th and 75th percentiles, respectively. The value for 'Whisker' corresponds to ±1.5 IQR (interquartile rate) and extends to the adjacent value, which is the most extreme data value that is not an outlier. The outliers are plotted individually as plus signs. **b** Median of mean fluorescent expression level as a function of the number of variants, for all variants containing each of the 41 motifs. Positive correlation indicates that up-regulating motifs are more likely to be detected in our experiment. The median value was calculated as the median of mean fluorescent expression level for each group of variants, containing one of the 41 motifs. **c** Pearson correlations of pairs of barcodes used to encode the same sURS. Correlations were computed by using the four normalized read bin values that were obtained for each barcode. **d** Averaging the Pearson correlation over a range of fluorescence values shows that a high correlation appears either in high- or low-expression variants supporting a likely regulatory function. Source data are provided as a Source data file.

distributions that are characterized by low median values and low number of variants are likely indicative of a repressive or inhibitory motif. The reduced number of variants in these low median expression level distributions is, therefore, likely due to strongly repressed variants that do not pass the SORT-seq assay's minimum fluorescence threshold (Supplementary Fig. 2). Conversely, highly expressing variants were collected by all bins, thus naturally leading to motif mean expression level distributions that are characterized by both a higher median and a larger number of variants that are, in turn, likely indicative of an up-regulating motif.

We next grouped all identical variant pairs (i.e., identical oligos with two different barcodes) and computed the correlation between the read-count 4-vector (corresponding to the number of reads in each of the four bins) obtained for each variant. We plot the correlation values obtained as a heatmap on an X–Y scatter plot, where the X and Y axes correspond to the mean fluorescent expression level of barcodes 1 and 2, respectively (Fig. 2c). The scatter plot shows variants exhibiting no correlation, complete anti-correlation, and fully correlated mean fluorescent expression levels (see color bar for Pearson correlations from −1 to +1). A closer examination of the plot reveals four identifiable correlation regimes: predominantly correlated pairs at low

expression (Expression (A.U.) < 2000) and high expression (Expression (A.U.) > 5000), predominantly uncorrelated pairs in the intermediate expression range (2000 < Expression (A.U.) < 5000), and an anticorrelated set of pairs. The anticorrelated pairs are likely due to various sources of noise, which include low read count, low number of viable yeast cells that were sorted, and variation in expression. The high correlation regions are consistent with up-regulating sURSs (high-fluorescence regime−Fig. 2c-top right) and down-regulating sURSs (low-fluorescence regime−Fig. 2c-bottom left), where the regulatory effect of the motifs strongly affects the expression. Finally, in the intermediate fluorescence regime, where the predominantly uncorrelated pairs are found (Fig. 2d-middle, between Expression (A.U.) of 1850 to 4150), the absence of strong regulatory effect may be a major cause for lack of correlation. For these weakly-regulating variants, small variation in sURS sequence due to the barcodes (for example) may have a dominant role in the expression output.

## Deep learning reveals that only variants with multiple barcodes generate statistically reliable data

To provide a more quantitative analysis of the data, we trained a convolutional neural network (CNN) to predict the gene expression level

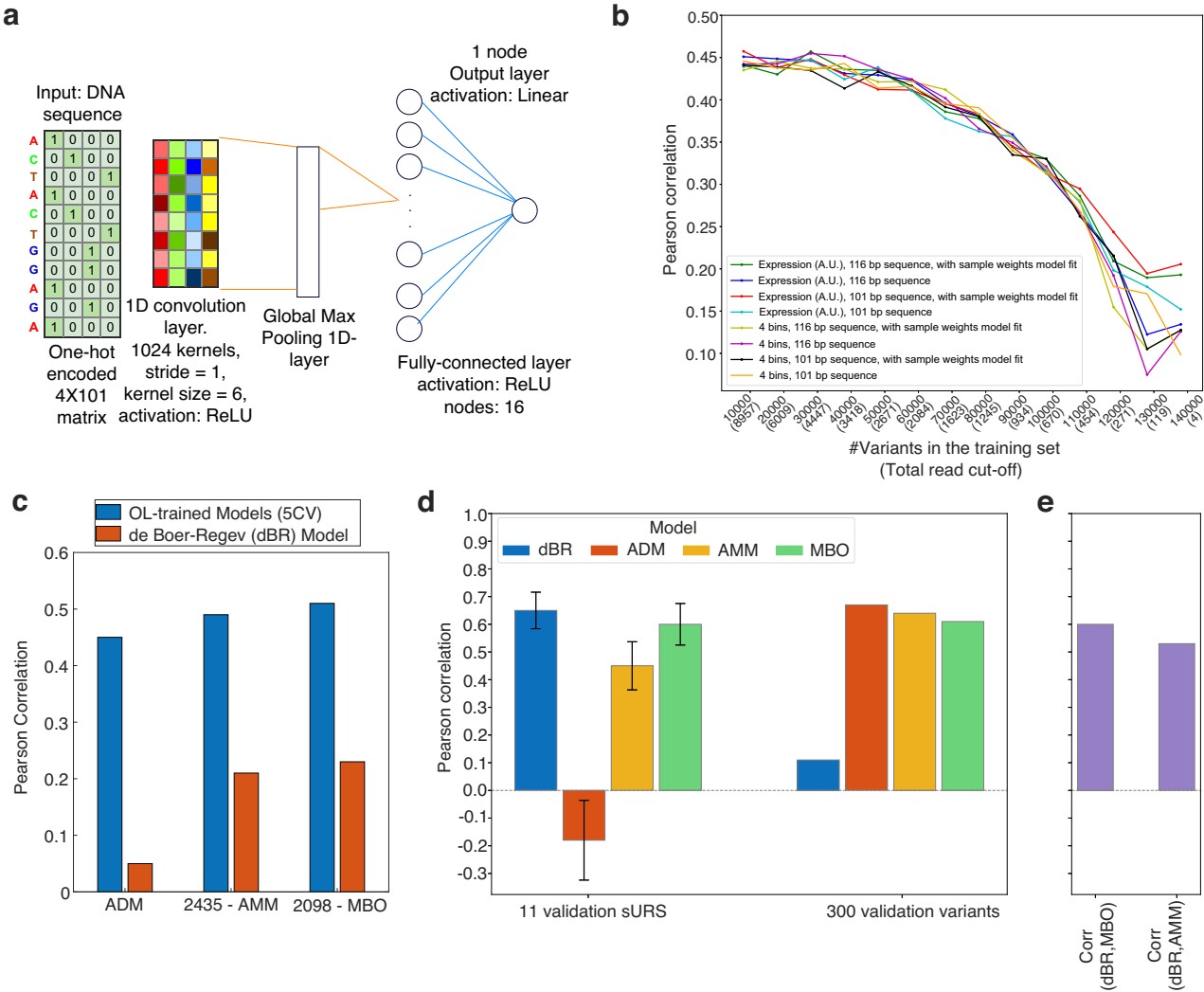

**Fig. 3 | Convolutional neural network identifies a set of 2435 variants with robust expression. a** Schematic of our convolutional neural network architecture. **b** Pearson correlation results for various models as a function of the training-set size sorted by variant total read counts. **c** Comparison of Pearson correlation results on a validation set for OL-trained models with that of the de Boer-Regev model. **d** Prediction performance comparison of the dBR, ADM, AMM, and MBO models on the 11 sURS variants validation set and the 300 variants with highest read count among the 2435 variants with 22 barcodes. Error bars were computed by bootstrapping which was carried out by sub-sampling all possible subsets of 9 of the 11 points and computing the Pearson correlation for each subset. **e** Pearson correlations between the predictions of the dBR model and the MBO and AMM models on the 11 sURS variants validation set. Source data are provided as a Source data file.

given the DNA sequence (Fig. 3a). First, we aimed to isolate the subset of variants in the OL for which robust and statistically reliable mean fluorescent expression levels were measured. Specifically, our goal was to identify the training set which yields the best regulatory predictive performance on a withheld test set of sURS variants from the OL. We tested 8 models, which differed by combinations of three characteristics: with/without a 15nt leader barcode sequence, variants characterized by a single mean fluorescent expression level or by a 4-vector corresponding to variant distribution over the 4 expression bins, and with/without considering their total read count in the training process (i.e., sample weight). Sample weights corresponded to the proportional contribution of the variant's total read count to the total read count of the library, or per bin, depending on the model. Each model was tested with multiple read cutoffs, varying from 8957 down to 4 total reads per variant. This in turn yielded an increasing number of variants by which the model was trained, ranging from 10,000 to 140,000, respectively. We tested the model on 2000 randomly held-out variants, selected from the 10,000 highest read-count variants. Altogether, 112 models were tested (for full details see "Methods").

The results of the Pearson correlation between predicted and measured mean fluorescence expression level of the held-out set are shown in Fig. 3b. In general, all models behave similarly: Pearson correlations are at their highest values when the models are trained on up to 20,000 sequences with a total read cut-off of 6009, and they decrease as the training set size increases (Fig. 3b). Interestingly, the prediction performance of the different models does not seem to be affected by including or excluding the barcodes, or by whether the variants are scored by mean fluorescent expression level or by the 4-bin distribution. Overall, the highest Pearson correlation achieved was a modest -0.45 at the plateau range, indicating that various sources of noise become increasingly dominant with a reduced read-count threshold. Consequently, we chose the 20,000 variants with total read count >6009 to be the training set for our initial "all-data" model (ADM).

Given the omni-present limiting noise in all models, we hypothesized that two dominant sources of noise may be the use of only two barcodes (the case for most OL variants: 146,802 variants), and a large subset of motifs that do not encode a regulatory function. This

prompted us to construct two additional models, based on the following training sets. The first model is based on the observation from the various ADM models that neglecting barcodes does not seem to affect the predictive capacity of the model, and is thus trained on the subset of 2435 variants with 22 total barcodes for each variant. We call this model the "all-motif" model (AMM), as it contains all variants with 22 designed barcodes. The second model uses a subset of 2098 variants with 22 barcodes for each variant and at least one mixed-base motif. This model is constructed based on the assumption that mixed-base motifs are more likely to be regulatorily active due the fact that they are composed of a larger number of potentially functional sub-variants. This model is referred to as the "mixed-base only" (MBO) model.

We then compared the performance of all three models on a held-out test set comprising 20% of the variants for each model. The Pearson correlations show a modest increase (Fig. 3c−blue bars) from the ADM baseline of 0.45 to approximately 0.48 for the AMM and 0.51 for the MBO model, despite the reduction by an order of magnitude in the size of the training set. Furthermore, we used the de Boer-Regev (dBR) model[24], trained on a different yeast expression library, to score the variants that make up the training set of each of our models. Here too (Fig. 2c−red bars), we observed a modest increase in the Pearson correlation between dBR-predicted and experimentally measured mean fluorescent expression levels from the ADM (0.05) to the AMM (0.21) and best-performing MBO (0.23) variant sets (see Supplementary Data 3 for MBO model score per variant).

To experimentally validate the predictions of the various models, we constructed 11 sURS variants containing an uncharacterized mix of three of our 41 motifs (Supplementary Data 6). The variants utilized consensus motif sequences without mixed bases and were chosen as follows: the desert variant and 10 additional variants with MBO-based predicted mean fluorescence expression levels spanning the expression range of 2700−3400 (A.U.). These variants were cloned individually upstream of the weak mCore1 promoter as before, integrated into the yeast genome, and their expression levels were measured via flow cytometry (see "Methods"). As an evaluation on a small set of only 11 variants is highly sensitive to small changes in prediction, we used bootstrapping to report robust correlations. We sampled all subsets of 9 variants out of the 11 and reported the average correlation over these subsets. Moreover, we held out as an additional test set of 300 variants that were supported by the most reads from the set of 2435 variants with 22 barcodes for each variant.

The prediction-performance evaluation on the 11 sURS dataset shows the dBR model outperforms both the AMM and ADM models (Fig. 3d). The dBR model achieves an average Pearson correlation of $0.65 \pm 0.066$, surpassing the AMM model with an average correlation of $0.45 \pm 0.087$, and significantly outperforming the ADM model, which achieved an average Pearson correlation of $-0.18 \pm 0.144$. However, the prediction-performance evaluation on the test set of 300 high-quality variants reveals that the ADM model performs better than both the AMM and dBR models. The ADM model achieved a Pearson correlation of 0.67, surpassing the AMM model with a Pearson correlation of 0.64, and significantly exceeding the dBR model, which achieved a Pearson correlation of 0.11. The MBO model achieved the best combined prediction performance on both test sets: an average Pearson correlation of $0.60 \pm 0.075$ on the 11 sURS variants, and a Pearson correlation of 0.61 on the 300 high-quality variants.

**Minimum-hyper-geometric analysis identifies activating and repressing motifs and sub-motifs from the 2435 variant subset**
We next proceeded to identify the motifs which likely play a regulatory role in our system by employing a minimum-hypergeometric (mHG) analysis[25] on the 2435-variant set (i.e., each variant encoded with 22 barcodes, see "Methods"). Based on this analysis, we computed the mHG p-value for enrichment of each of the 41 motifs and the desert

motif (Fig. 4a and Supplementary Fig. 3). We identified 7 motifs that were enriched at the top of the list (blue bars−mHG p-values $< 10^{-4}$), 6 motifs that were enriched at the bottom of the list (red bars−mHG p-values $< 10^{-4}$), and the remainder of the motifs were either unenriched or weakly enriched. The $10^{-4}$ enrichment limit was set based on the results of the desert motif (black bar) and the apparent empirical jump in p-values at the top of the list between $10^{-2}$ and $10^{-4}$. In the inset, we present mean fluorescent expression level boxplots of all variants containing selected top-enriched (activating) and bottom-enriched (repressing) motifs, and compared to the rest of the variants, showing a clear up-shift or down-shift in the distributions that aligns with the mHG p-value computation. Of the 13 identified significantly enriched motifs, 8 were mixed-base (5 of 7, and 3 of 6, top and bottom of the list, respectively) and 5 were of the non-mixed variety, providing further validation for the success of the MBO model.

Next, we applied mHG analysis to identify enriched sub-motifs within both the enriched and unenriched mixed-base motifs. To do so, we evaluated separately the number of reads and mean fluorescent expression level for each variant containing a particular sub-motif and reconstructed the mHG distributions in accordance with the new sub-variants (see "Methods" for details) (Fig. 4b−e). In each panel, the motif boxplots as compared with the rest of the population are shown on the left, while on the right we show the split sub-variant distribution, which can vary from 2 (e.g, panel b) to 16 (see Supplementary Fig. 4 for all sub-variant distributions). The panels show that in some cases, only one of the sub-motifs is enriched (Fig. 4b−d), while in others all sub-motifs are enriched but with different p-values, suggesting a varying degree of enrichment (Fig. 4e). Altogether, we identified 5 enriched non-mixed-base motifs and 13 enriched sub-motifs (mHG p-value $< 10^{-4}$ see Supplementary Data 4 and 5, and Supplementary Figs. 3 and 4), which can be considered likely candidates for functional TFBSs. Of these functional motifs, 8 motifs are associated with down-regulation, while 10 are associated with up-regulation. Interestingly, 5 out of the 18 functional motifs do not have a known associated TF in any organism[21].

**Boosting expression with statistically significant motifs that are embedded within sURS**
Based on the mHG analysis, we chose 23 motifs and sub-motifs for validation experiments: 20 motifs and sub-motifs with p-value < 0.001, and 3 sub-motifs with higher p-values for sub-motif comparison (up to p-value of 0.01) (see Supplementary Data 4 and 5). In addition, we designed a longer desert regulatory sequence in silico, in a similar fashion as we did previously for the OL desert (see "Methods"). This enabled testing of the regulatory effect of up to six DNA motifs, compared to the OL variants, which incorporated up to three motifs per variant (see Supplementary Data 1). We designed the validation oligos (see Supplementary Data 6) to address the following: 1. Evaluate the regulatory effect of weaker, but potentially significant, motifs ($10^{-4} <$ mHG p-value $< 10^{-3}$), 2. Validate new and strongly activating/repressing motifs individually (p-value $< 10^{-4}$), and 3. Examine the effect of the positions of the motifs within the regulatory sequence on expression levels. The results (Fig. 5a−c) show that sURSs behave as predicted: cassettes containing predicted up- and down-regulatory motifs boost and inhibit expression, respectively. Up- and down-regulation were observed for both promoter backgrounds, suggesting that the total expression level is a multiplicative product of the core-promoter expression level and the degree of regulation generated by the sURS. In particular, boost values of x3-x10 and x2-x4 were observed for a weak and a strong core promoter, respectively, while approximately x1.5-x4 inhibition of expression was observed for both promoter backgrounds. Also, in general, no major position-dependent effect was detected, (see Fig. 5b, c).

To determine whether the effect of the motifs is additive as predicted by the billboard model[26], we inspected the fold-regulation

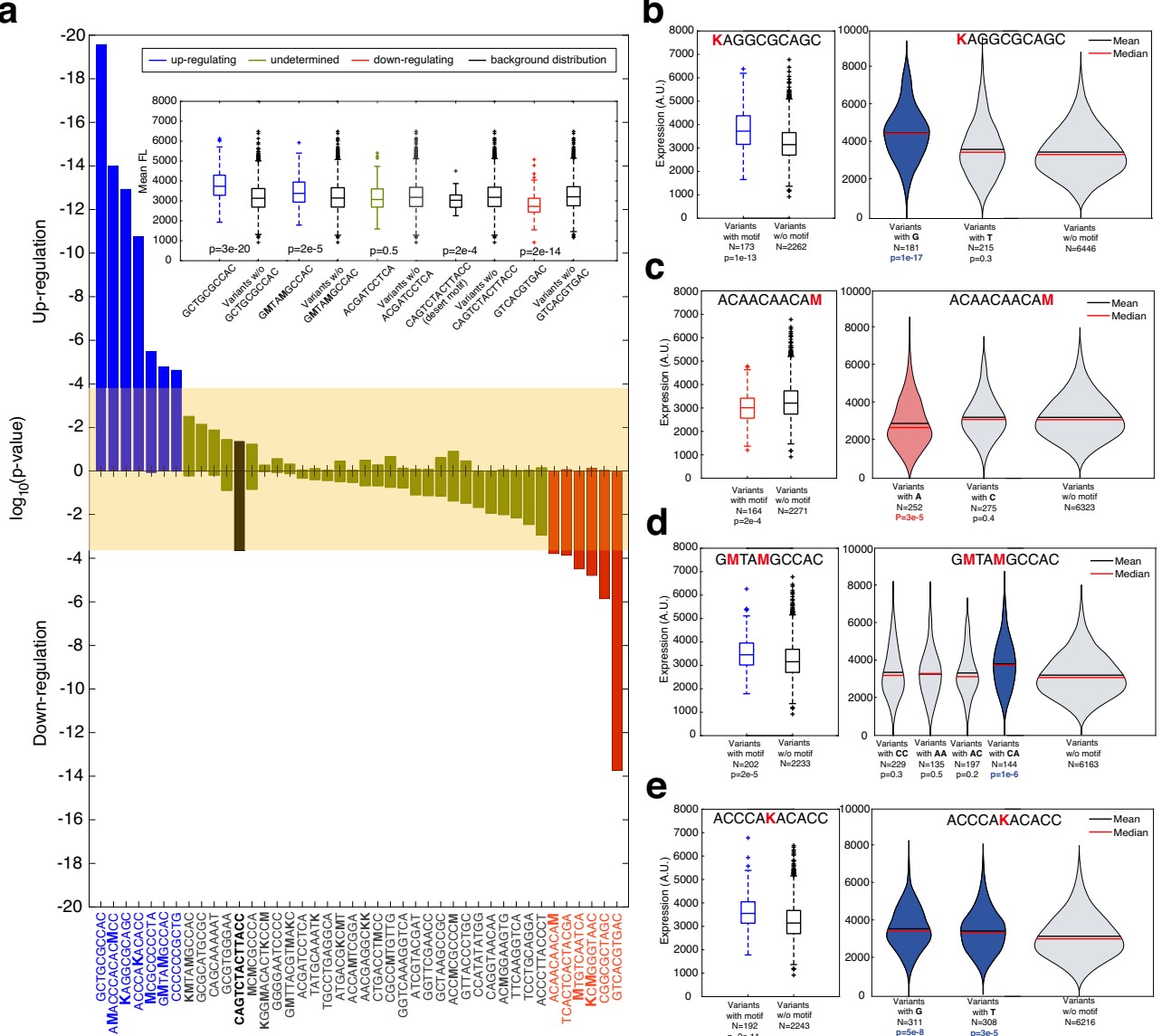

**Fig. 4 | mHG analysis on the 2435-variant set reveals functional up- and down-regulating motifs. a** *p*-Values resulting from mHG analysis for each motif for variant lists arranged in order of decreasing expression levels (up-regulating) and increasing expression levels (down-regulating). A cutoff of *p*-value < $10^{-4}$ was chosen for determining up-(blue) and down-(red)regulating motifs. The *p*-values for the desert motif are labeled in black, and for undetermined motifs in green. (Inset) Boxplots for a select set of variants ($n_w$) containing a particular motif compared with the rest of the variant population not containing the motif ($n_{wo}$). (Left) $n_w = 168$ and $n_{wo} = 2267$ variants (middle-left) $n_w = 202$ and $n_{wo} = 2233$ variants (middle) $n_w = 167$ and $n_{wo} = 2268$ variants (middle-right) $n_w = 58$ and $n_{wo} = 2377$ variants (right) $n_w = 163$ and $n_{wo} = 2272$ variants. **b–e** Sub-motif analysis for a select set of

motifs. (Right) Box-plot comparison for the inset for the full-mixed-base motif. (Left) Violin plots for all variants containing the various sub-motifs compared with all sub-variants that do not contain the sub-motif. Blue-shaded violin plots are sub-motifs that were determined to be significantly activating according to mHG analysis. Red-shaded violin plots are sub-motifs that were determined to be significantly repressing, based on the analysis. On each box, the central mark indicates the median, and the bottom and top edges of the box indicate the 25th and 75th percentiles, respectively. The value for 'Whisker' corresponds to ±1.5 IQR (interquartile rate) and extends to the adjacent value, which is the most extreme data value that is not an outlier. The outliers are plotted individually as plus signs. Source data are provided as a Source data file.

measurement as a function of the mean $\log_{10}$(*p*-value) that was previously determined by mHG analysis for all the motifs in the 6-motif cassettes (Fig. 5d). Here, we assume that the mHG *p*-value is a good experimental proxy for the degree of fold regulation, as was shown in Fig. 4a. The plot shows that for 6-motif cassettes, whose fold-regulation was measured over the weak promoter background (blue squares), a distinct linear dependence on the mean $\log_{10}$(*p*-value) is observed (Pearson correlation >0.85). This finding supports a linear additive regulatory effect, where the validated motifs have no interaction with one another. By contrast, for the down-regulating 6-motif cassettes measured over the strong-promoter (red circles), no additive

effect was observed. Instead, a constant -x2 fold down-regulation is observed, independent of mean $\log_{10}$(*p*-value) computed for each cassette. Finally, we compared the mean fold boost and attenuation effects for the 6- and 3-motif cassettes (Fig. 5e). For boost (blue bars), a clear dependence on the number of motifs is observed, while for attenuation (red bars), no such dependence is detected. Taken together, our mHG analysis and validation experiments support a regulatory model (Fig. 5g), where the expression level can be quantified as a product of the core expression level (Fig. 5f), the cumulative total boost contribution of all motifs on the sURS, and a -x2 inhibition if one or more attenuating motifs are present. In our experiment and with

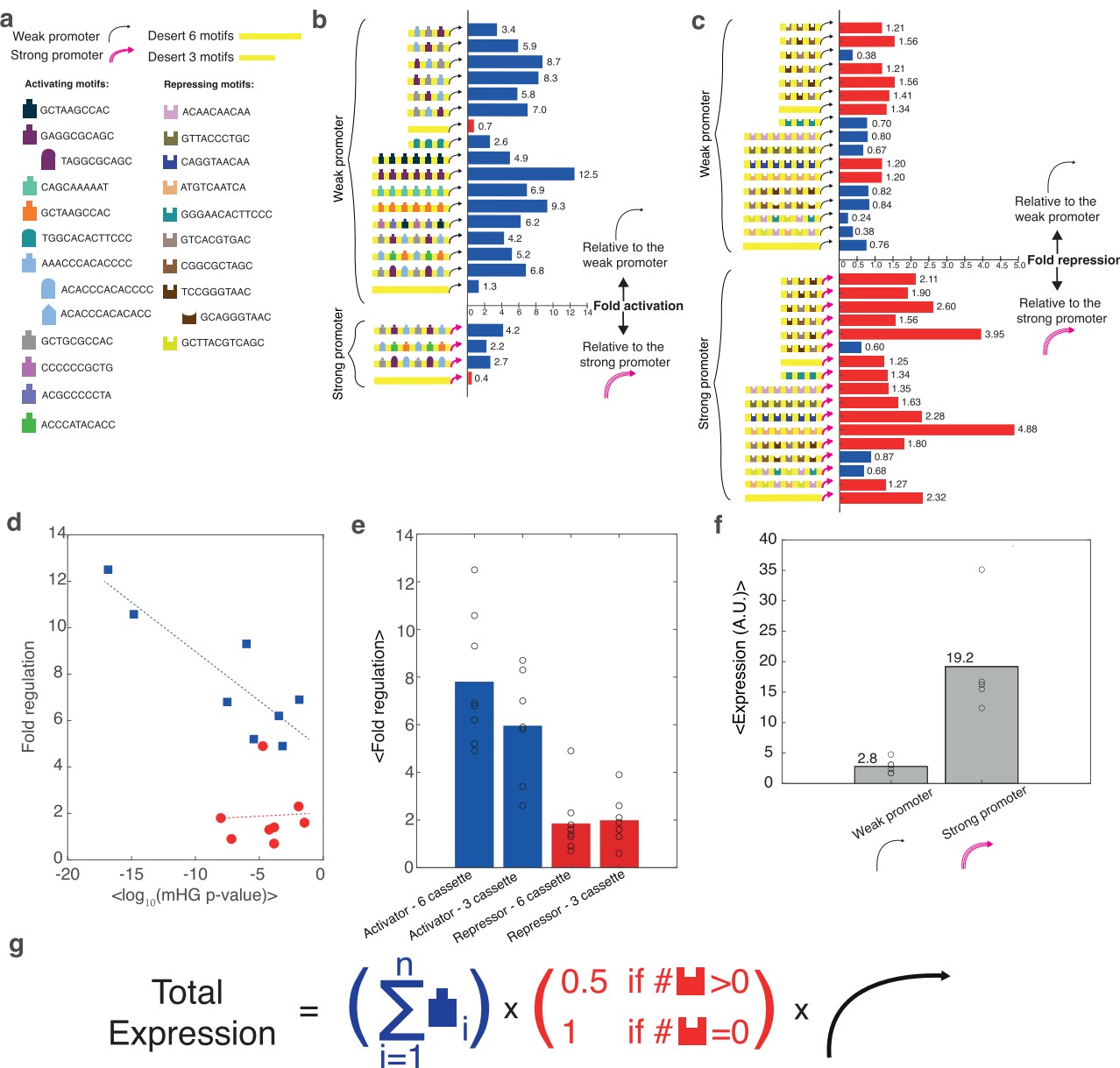

**Fig. 5 | Single-clone experiments in yeast validate mHG predictions for motifs.** **a** A legend depicting all the different parts in the validation variants: promoters, desert sequences, and various up- and down-regulating sub-motifs. **b, c** Fold regulation (activation and repression, respectively), relative to weak (upper plot) and strong (lower plot) promoters, shown as bar plots of various single clones as depicted by the promoter and URS schema next to each plot. In blue—activating variants, in red—repressing variants. **d** Fold regulation of activating motifs (in blue) and repressing motifs (in red) as a function of the log of *p*-value computed by the mHG analysis. **e** The average of fold regulation values for different regulatory architectures depending on the type of regulatory motifs and the number of motifs in the cassettes. (Left) mean up-regulation for $n = 8$ 6-motif sURS cassettes. (Middle-left) mean up-regulation for $n = 7$ 3-motif sURS cassettes. (Middle-right) mean down-regulation for $n = 8$ 6-motif sURS cassettes. (Right) mean down-regulation for $n = 7$ 3-motif sURS cassettes. **f** The average of the median fluorescent expression level for the weak ($n = 4$) and strong promoters ($n = 4$) devoid of any added URSs. **g** A proposed expression model for the synthetic yeast promoters depicting the total expression level of a gene given its regulatory architecture. This is based on: the number of activating motifs, the presence/absence of repressing motifs, and the core promoter. Source data are provided as a Source data file.

these characterized motifs, this "motif additive model" (MAM) yielded a library of verified sURSs in yeast spanning ~x50 range in expression level of yeCitrine with the same core promoter.

## Boosted expression level translates from yeast and mammalian cells

The model shown in Fig. 5g indicates that for a given core promoter, gene expression is a product of independently functioning cis-regulatory elements and the core promoter strength. Since the motifs used in our experiments are predominantly conserved across organisms, the MAM model suggests that the predictions generated

thus far should also be valid for any growth condition in yeast and in mammalian cells if the biological role of the TFs that bind these DNA motifs is also conserved in higher eukaryotes across genomes.

To test this assertion, we first tested the 43 sURS variants that were synthesized to validate the MBO model (11 variants—Fig. 3d) and the mHG analysis (32 variants—Fig. 5) in three additional growth conditions: SD-Ura/mCore promoter/2%-glycerol/30 °C, SD-Ura/mCore-promoter/2%-glucose/39 °C, and SD-Ura/mCore promoter/2%-glucose/30 °C/1 M NaCl. Together with the two conditions tested in Fig. 5 (i.e. SD-Ura/mCore promoter/2%-glucose/30 °C and SD-Ura/FEC-mCore promoter/2%-glucose/30 °C), we were able to compare the

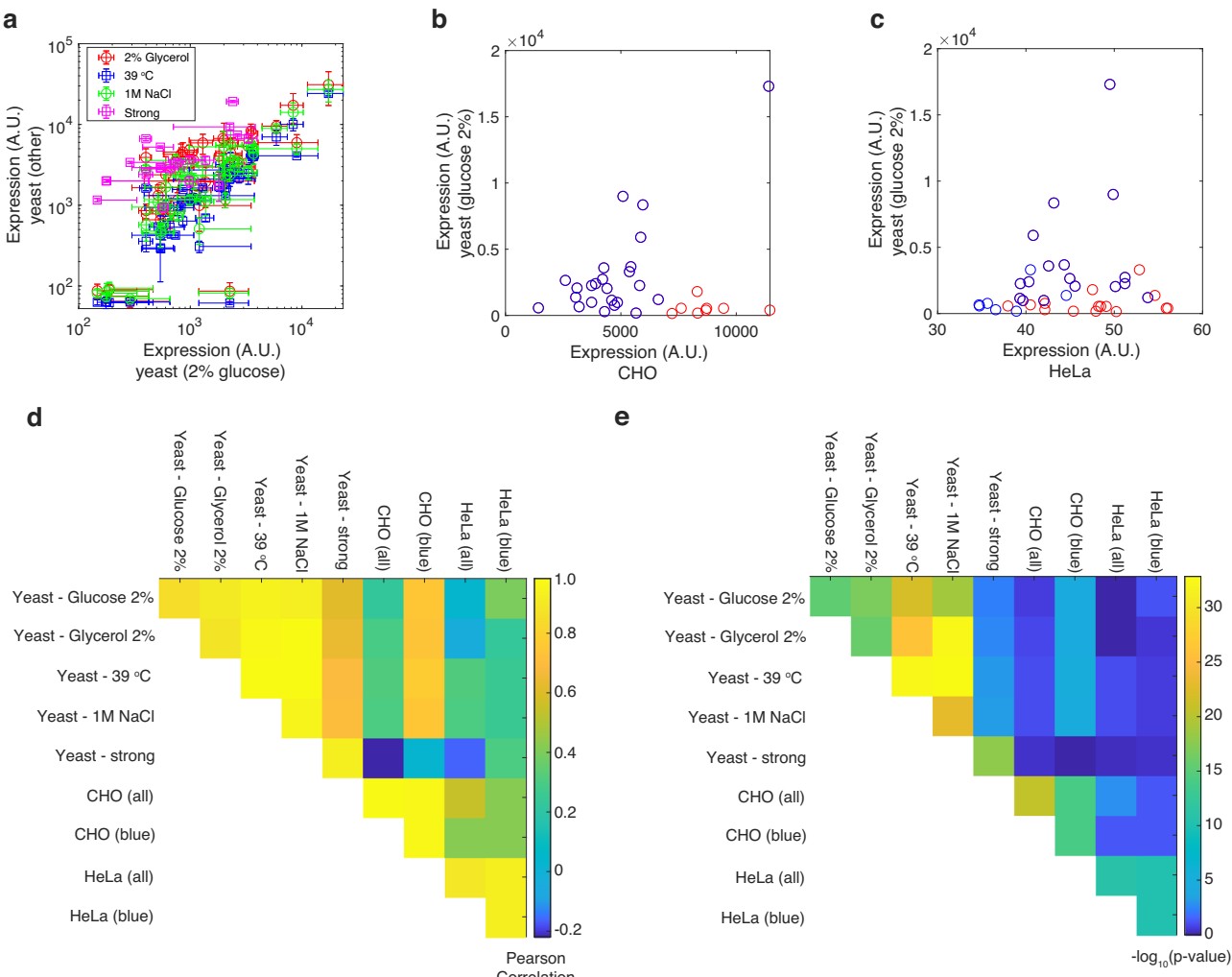

**Fig. 6 | sURS variants function similarly in different cell types and growth conditions. a** Comparison of sURS variants expression levels in yeast. Growth conditions are as follows: glucose 2% refers to SD-Ura+2% glucose, 30 °C, weak core promoter (mCore), no additives, 43 variants. Glycerol 2% refers to SD-Ura+2% glycerol, 30 °C, weak core promoter (mCore), no additives, 43 variants. 1 M NaCl refers to SD-Ura+2% glucose, 30 °C, weak core promoter (mCore), 1 M NaCl added, 43 variants. 39 °C refers to SD-Ura+2% glucose, 39 °C, weak core promoter (mCore), no additives, 43 variants. Strong refers to SD-Ura+2% glucose, 30 °C, strong core promoter (FEC-mCore), no additives, 20 variants. Error bars were computed using standard-error analysis carried out on mean flow cytometry fluorescence measurements obtained from $n = 3$ biological repeats. **b**, **c** Comparison of the regulatory output for 32 sURS variants in CHO (**b**) and HeLa (**c**) cells to the regulatory output obtained for the 2% glucose growth condition in yeast. Red circles correspond to strongly repressed yeast variants that express strongly in mammalian cells. **d** Pearson correlation coefficients obtained for each pair of conditions. **e** Pearson correlation $p$-values obtained for each pair of conditions. Source data are provided as a Source data file.

results of four separate growth conditions and two promoter architectures. For each condition and architecture, measurements were carried out in biological triplicates (see Supplementary Fig. 5) and growth curves were assessed (see Supplementary Fig. 6) to ensure that there were no adverse effects generated by the sURS sequences on the growth. The results show (Fig. 6a) a tightly correlated set of expression levels for all five conditions supporting regulatory roles of the sURS sequences that are independent of growth conditions and core promoter architecture in yeast.

Next, to test the broader applicability of the sURS sequences, we created for mammalian cells a regulatory architecture similar to the one synthesized for yeast that consisted of an sURS sequence positioned immediately upstream of the minimally active mammalian promoter pCMV$_{min}$ and an mCherry reporter. We created 32 mammalian sURS variants containing the 3 and 6 motifs that we used previously to validate the mHG analysis (see Fig. 5 and Supplementary Data 6). As a positive control for mCherry expression, we used the full pCMV promoter.

We first transfected the mammalian variants (encoded on a plasmid) to CHO-K1 MI-HAC cells (hereby referred to as CHO) and assayed mCherry reporter expression using flow cytometry. We compared the mean fluorescence expression levels in CHO cells directly to the levels obtained for the same yeast variants grown in 2% glucose (Fig. 6b). The data shows a relatively correlated group of variants (blue), and an uncorrelated group of variants (red). The 24 "blue" variants that correlated well generated a statistically significant Pearson correlation of 0.75 ($p$-value 2.41e-5). The uncorrelated "red" variants were all various permutations of three motifs that were found to be repressive in yeast (GCTGCGCCAC GAGGCGCAGC AAACCCAC ACCCC). This means that one or more of these motifs is likely activating in CHO cells due to the high mean expression observed from the sURSs. Next, we tested our variants in HeLa cells, and also compared the results to the 2% glucose yeast sample (Fig. 6c). The results show a similar relationship to yeast expression as was observed for CHO cells with a Pearson Correlation of 0.41 ($p$-value 0.054) for the same blue and red grouping of variants.

To further test for the cross-species applicability of these validation variants, we computed the correlation coefficient (Fig. 6d) and $p$-value (Fig. 6e) for any pair of datasets of the seven tested. The plots show that the regulatory output of the sURS variants in yeast are highly correlated across all five yeast conditions with a high degree of statistical significance ($p$-value < 1e-10). For the mammalian variants, the correlation between the HeLa and CHO datasets was similar to what was obtained by correlating the 24 blue CHO variants (i.e., blue circles in Fig. 6b) and the various yeast conditions, yielding moderately significant $p$-values that ranged between 1e-3 to 1e-5 (see Supplementary Data 7 for precise values). The correlation for blue HeLa variants with yeast was somewhat lower than what was obtained for CHO but was nevertheless marginally statistically significant for the 2% glucose growth condition ($p$-value ~ 0.05). The similar cross-species correlations of yeast-CHO, yeast-HeLa, and CHO-HeLa with moderate statistical significance provide support to an interpretation that 24 of the 32 sURS sequences generate a regulatory response that is independent of the eukaryotic cell-type that was used.

## Modeling expression for all cell-types using a combined machine-learning and mechanistic model

To provide further support for the cell-type-independent interpretation of the experimental data, we hypothesized that an ML model trained on yeast data should also provide adequate predictions for the mammalian expression data. Thus far, in this work, we have developed two models: the MBO model that was trained on the 2435-variant set, and a mechanistic motif additive model (MAM - see Fig. 5g) that was developed based on the mHG analysis. Both models are deficient for the above stated task. The MBO model cannot be applied to the 6-motif variants, as it can only provide reliable predictions for 3-motif sequences that are identical in length to what it was trained on. The mechanistic MAM model is based on a simplified first-order model of transcriptional regulation, which assumes that the regulatory output is proportional to the sum of the individual contributions of each functional motif. Consequently, we needed to construct a hybrid ML-additive model (MLAM) that on the one hand can be applied to an sURS containing any number of motifs, while on the other hand, properly models second-order sequence-level events that are captured by the MBO model and neglected by the MAM model.

To generate the MLAM model, we extended our CNN by an additional input of a 41-long motif count vector. This additional input is provided by concatenating it to the output of the pooling layer to enable combination of both sequence-level features and motif-occurrence counts (see Supplementary Fig. 7 for schema). In addition, we expanded the input to the maximum length and trained on shorter sequences by padding them first. We tested the MLAM model by providing a prediction for every single variant of the 43-variant validation set. First, we compared the prediction performance of the MAM to the MLAM model for all five yeast conditions tested. The results show (Fig. 7a) that the MLAM model improved prediction performance over all five datasets indicating that it captures not only the first-order mechanistic effects, but also second-order sequence-level effects to improve its reliability. We then tested both models on the 24 "blue" mammalian variants that experimentally correlated well with yeast. The results shows that while the MAM model failed to generate a reliable prediction for both the CHO and HeLa cells, the MLAM model was able to generate predictions that were similarly reliable for both yeast (Fig. 7b) and mammalian (Fig. 7c) cells.

## Discussion

We provide a synthetic biology design algorithm called UNILIB for the generation of a non-inducible boost of gene expression in yeast and mammalian cells. UNILIB is comprised of two components: a sequence generator for a synthetic upstream regulatory sequence (sURS) consisting of multiple motifs characterized in this study, and the MLAM computational model that provides a reliable prediction for the sURS regulatory output. The UNILIB algorithm was developed through an OL-ML study, and was validated on 43 unseen sURS sequences in yeast and mammalian cells. The UNILIB boost can be modeled, to a first-order approximation, as a cumulative sum of the individual boosts of the motifs embedded within the sURS that were empirically determined using the large-scale OL experiment. The UNILIB algorithm, therefore, can be utilized as a component in a design tool for protein over-expression in non-bacterial cells. Specifically, we show boosts of gene expression levels of >1000% and >400% in validation experiments for generic weak and strong promoters, respectively, in yeast providing a proof-of-concept of the potential utility of UNILIB. In addition, UNILIB can simplify synthetic gene regulation by reducing the dependence on external components (e.g., synthetic TFs, inducers, etc.)[5,17]. Furthermore, UNILIB can also allow for a more nuanced control of gene expression, thus facilitating the implementation of more complex gene circuits that require a modular control of protein levels. The nuanced control can be generated by a fine-tuned design which incorporates both boosting and attenuating motifs characterized in this study, provided that a suitably functional core promoter is selected to match the UNILIB sURS. Such nuanced control can be critical for more complex fermentation processes that involve optimized expression of genetic circuits for the bio-production of some industrially relevant chemicals or raw material. Together, both the computational and experimental tools developed in this work provide a constitutive promoter-design resource to the alternative-protein, synthetic-biology, and broader life-sciences communities. These tools, either together or separately, will allow users to devise various functional synthetic high, middle, and low expressing promoters, thus substantially shortening the design-build-test-learn cycle of synthetic regulatory systems in eukaryotes[27,28].

To develop UNILIB, we expanded the OL-ML approach by showing that PWMs can be encoded and characterized within a synthetic OL context. We employed a synthesis approach which allowed us to introduce K (G/T mix) and M (C/A mix) at selected motif positions that align with the frequency of these nucleotides in the PWM logo. Our OL-data and follow-up validation experiments show that encoding mixed-base motifs discovered in other organisms is likely to yield functional regulatory sequences (Table 1). 11 of the 20 mixed-base motifs were found to be functional, with 8 being identified in an initial all-motif analysis, and an additional 3 motifs identified in a refined sub-motif analysis. This result stands in contrast to the yield of only 5 functional motifs from the 21 non-mixed-based motifs. Consequently, we conclude that encoding PWMs, even in the simplified format employed here, is likely to yield wide-ranging and important regulatory results.

The specific motifs that we detected are categorized into four main groups. The first group is composed of previously unknown motifs. Altogether, we characterized 5 unknown motifs as either up- or down-regulating. For instance, GAGGCGCAGC motif was an unknown motif from *S. cerevisiae*, and through the mHG analysis and the validation experiments, we discovered that this motif is strongly boosting. The second group is composed of validated known motifs. For example, the Zic3 transcriptional activator, which is expressed in murine ES cells, binds the CCCCCGCTG motif. This motif was found to be boosting also in *S. cerevisiae*. The third group includes motifs that according to the literature are bound by dual-regulatory TFs. For example, the AMACCCACACMCC motif (sub-motifs: MM = AC/CC/CA) from *S. cerevisiae* is the binding site for Rap1 (Repressor-activator protein). Rap1 is associated with both up- and down-regulation in various yeast promoters[29,30]. Our analysis shows that given the growth conditions and setup used in our experiments and the architecture of our synthetic promoters, two of the AMACCCACACMCC sub-motifs are boosting motifs, with the 'AC' sub-motif being particularly strong. The last group consists of contradictory motifs, namely known repressing motifs that were found to be boosting, and vice versa. For

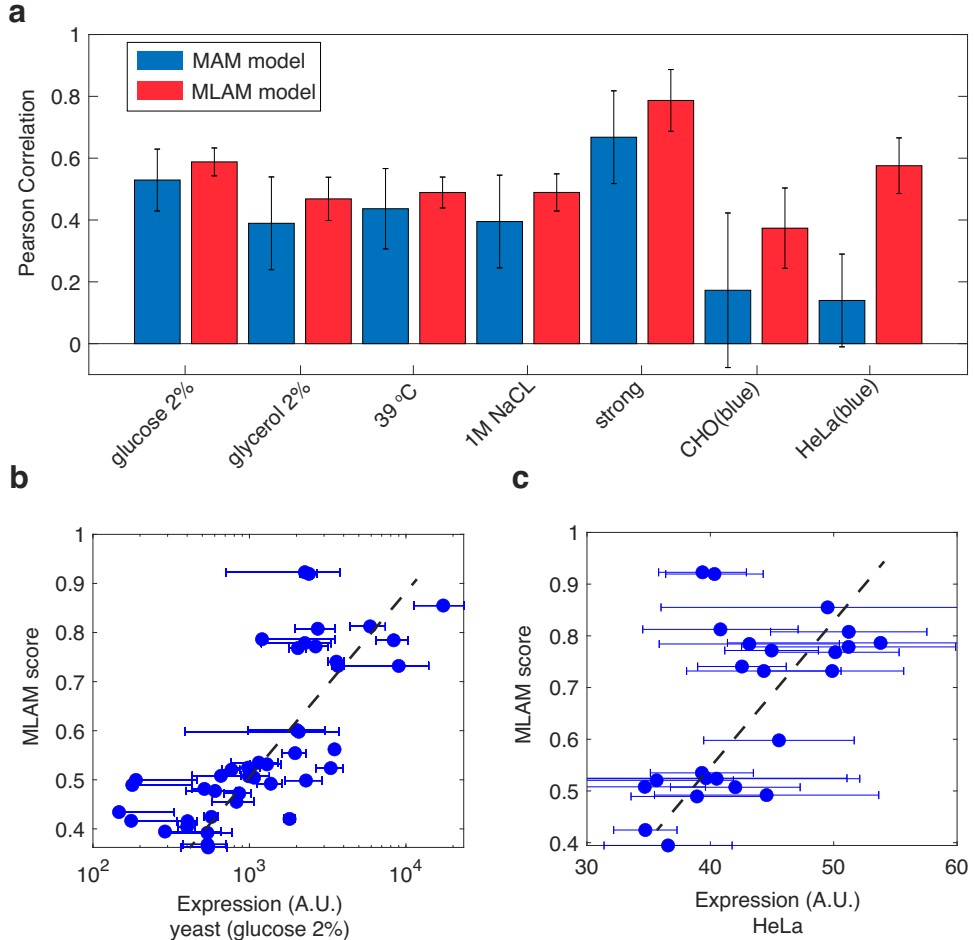

**Fig. 7 | A hybrid machine-learning and additive model (MLAM) predicts regulatory output in different cell types and growth conditions. a** Bar plot comparing the performance of the MAM (blue) and MLAM (red) models on all 7 experimental datasets. Error bars were computed by bootstrapping which was carried out by sub-sampling 100 times 30 of the points in each dataset (16 points for the mammalian datasets) randomly and computing the Pearson correlation for each subset. **b, c** Plots depicting the MLAM scores vs the 2% glucose (**b**) and HeLa (**c**) fluorescence measurements. Both datasets show a linear relationship between the fluorescence expression data and the MLAM scores consistent with the high Pearson correlation coefficient. Error bars were computed using standard-error analysis carried out on mean flow cytometry fluorescence measurements obtained from $n = 3$ biological repeats. Source data are provided as a Source data file.

**Table 1 | Motif sequences and their respective regulatory functions in the yeast *S. cerevisiae***

| Motif sequence | Regulatory function from literature (TF- if known, organism) | Characterized regulatory function based on this study |
|---|---|---|
| GCTGCGCCAC | Activating (Prz1, *S. pombe*) | Activating |
| CCCCCCGCTG | Activating (Zic3, Mouse) | Activating |
| GTCACGTGAC | Dual function (bHLH, Mouse and yeast) | Repressing |
| CGGCGCTAGC | Activating (STP, *S. cerevisiae*) | Repressing |
| TCACTCACTACGA | Activating (Abf1, *S. cerevisiae*) | Repressing |
| GGGAACACTTCCC | Activating (Nfi, Mouse) | Repressing |
| TCCGGGTAAC | Dual function (Reb1, *S. cerevisiae*) | Repressing |
| ACGCCCCCTA | Dual function (CTCF, Mouse) | Activating |
| ACCCAGACACC/ACCCATACACC | Dual function (Rap1, *S. pombe*) | Activating |
| AAACCCACACCCC/ACACCCACACACC | Dual function (Rap1, *S. cerevisiae*) | Activating |
| GCTTACGTCAGC | Dual function (Yap, *S. cerevisiae*) | Repressing |
| GAGGCGCAGC | Unknown (*S. cerevisiae*) | Activating |
| ACAACAACAA | Unknown (*D. melanogaster*) | Repressing |
| GCTAAGCCAC | Unknown (*S. pombe*) | Activating |
| GCTAAGCCAC | Unknown (*S. cerevisiae*) | Activating |
| ATGTCAATCA | Unknown (Mouse) | Repressing |

example, the ACGCCCCCTA motif is known to bind the CTCF repressor in murine ES cells. In our analysis, we found that it acts as a boosting motif in *S. cerevisiae*. Also, another surprising example is the CGGCGCTAGC motif, which is the binding site of Stp in *S. cerevisiae*. Stp is classified as an activator, regulating amino acid permease genes[31], and in our analysis it was found unexpectedly to be an attenuating motif in *S. cerevisiae*.

Our results provide further insight into the underlying eukaryotic regulatory grammar. First, we provide support for a version of the billboard model for transcriptional regulation, where different cis-regulatory elements (i.e., core promoter, activating URS, and repressing URS) each contribute to the output expression level in a manner that is independent of the other elements[32]. In particular, sURS sequences can either boost or attenuate expression in a manner that is directly proportional to the sURS motif content. Second, we show that regulatory rules associated with expression boosting and attenuation differ. While boosting was observed to be additive in our experiments, attenuation or inhibition of expression was manifested via a global scale-down of expression independent of the number of attenuating binding sites on the sURS. Third, the convergence of both the MBO and the de Boer model[24], which were trained on different experimentally generated promoter architectures, indicate that there is a large amount of redundancy in transcriptional regulation. Our work, the de Boer study[24], and others[8,33] further suggest that most mutations in the regulatory region may be non-essential (aside from fine-tuning expression at the local level), and thus do not radically affect the efficacy of the ML-based prediction models on unseen sequences, in different growth conditions, or in different cell types. This observation is further supported by a lack of detectible interaction between motifs, and the independence of the mean expression level from the position of the motifs within the sURS (see Fig. 5 and Supplementary Fig. 8). Together, these grammar rules provide an explanation as to why 24 of 32 "unseen" sURSs that were used for validation of the mHG-linear model in yeast, functioned in a similar fashion in CHO and HeLa cells. Consequently, these findings support a supposition that at least some of the eukaryotic regulatory genomic code is not as complex as previously thought, opening the door to constructing additional design algorithms that will further boost gene expression levels leading to a universal over-expression capability in non-bacterial cells.

## Methods

### Desert upstream regulating sequence design
We designed a 101 bp-long synthetic sequence, which lacks any known binding sites for yeast TFs in silico (for sequence, see first row in Supplementary Data 1). The aim was to design a sequence with a characterized basal level expression that maximizes the probability that only the designed TFs bind (given that their TFBS motifs are incorporated into the desert sequence). This so-called "desert" sURS was used as either the spacer sequence between motifs, or as a no-motif control. To generate this sequence, we used the IUPAC motifs deposited in "The Yeast Transcription Factor Specificity Compendium" (YetFaSCo) database (version 1.02)[22].

The desert sequence was generated by avoiding YetFaSCo TFBSs with the following realistic limitations: 1. Short motifs (fewer than 5 nucleotides) were eliminated from the screen due to their high probability to appear in the designed sequence. 2. All lower-case motifs in the database were eliminated. According to the IUPAC code, lower-case nucleotides indicate lower frequency. 3. Motifs containing 5 or less upper-case characters were removed from the database regardless of their length.

### Motif selection and encoding into the OL
We selected motifs for the OL from a list created by an empirical assay designed to test protein activity in a broad swath of organisms[21]. That study focused on developing a protein activity assay specifically for TFs in cell and tissue extracts. Based on the study's data, we selected 41 enriched motifs from various organisms and tissue cells (e.g., yeast, fly, and mouse tissue cells), according to the following criteria: (i) different motif types: 8 organism-shared motifs, 5 mouse tissue-shared motifs, 14 unshared motifs (unique to an organism), and 14 unknown motifs (with unknown regulatory function); (ii) known/unknown regulatory function: 27 of the selected motifs have known regulatory function and thus were anticipated to be the control motifs for the 14 unknown motifs that we wanted to characterize.

Motifs were encoded via a mixed-based approach using Twist Bioscience novel DNA synthesis technology. Additional complexity in the designed library, apart from the motif-shuffled variants, were the K (G/T) and M (A/C) mixed bases, incorporated in 20 motifs at pre-determined positions in the motifs according to their PWMs. K and M substitutions were based on the percentage calculation of the respective G/T and A/C occurrences in each position of the motif[21]. Positions within the motif with dominant percentages were replaced by either K or M in the final design. 70% threshold was set to determine the K/M substitution, but the actual calculated threshold was higher at 73% (see "Results" section).

### Design of motif-based sURS OL
The designed motif-based sURS OL includes the following DNA parts: a unique 15nt barcode sequence for each oligo in the OL and a 101 nt-long variable region, which is based on the designed desert sURS chassis. Specifically, barcodes were designed with a minimal 3 nt Edit distance and restricted to maximum 3 nt of sequentially repeated bases with GC content between 35% to 65%. Both ends of each oligo were flanked with restriction sites (SpeI and EagI) for library cloning purposes and forward/reverse primer sequences for the library's PCR amplifications. Motifs from the set of 41 TFBS motifs were placed at fixed positions along the desert sequence depending on the number of motifs placed in the sequence (single/double/triple-motif oligos). The motif positions were at the 46th base for single-motif variants, at the 28th and 66th bases for double-motif variants, and at the 18th, 46th and 74th bases for triple-motif variants. In addition, motif sequences were "shuffled" in the desert chassis (in accordance with the motif placement guidelines mentioned above) resulting in 70644 different variants, including the desert variant lacking the 41 motifs. All variants were represented by 2 barcodes, except 2435 variants, picked randomly from the pool, that had an additional 20 barcodes (22 barcodes in total). In total, 189,990 barcoded oligos were designed (for sequences, see Supplementary Data 1). The library was synthesized by Twist Bioscience.

### Cloning the OL plasmid
We used a p416-based plasmid with a synthetic truncated core promoter, developed by Redden and Alper[23], to clone the OL upstream to the minimal core1 (mCore1) promoter. The mCore1 promoter has a weak activity and regulates the fluorescent yeast-enhanced Citrine (yeCitrine) gene. The plasmid has *URA3* as a yeast selection marker and lacks the cen/ars sequence, enabling plasmid integration into the yeast genome at *URA3* locus. To prepare the plasmid for the OL cloning, we inserted a short duplexed DNA (made by standard oligo hybridization), containing SpeI and EagI restriction sites, into the plasmid upstream of the mCore1 region (for primer sequences see Supplementary Data 1). Initially, the plasmid was digested with AscI (NEB #R0558) and treated with CIP (NEB #M0290) to prevent self-ligation. Then, the mCore1 plasmid and the short-duplexed DNA were ligated at 1:3 ratio by T4 DNA ligase (NEB #M0202). We then transformed ligated plasmid to *E. coli* TOP10 chemically competent bacterial cells (Thermo Fisher Scientific), and a positive clone was verified by Sanger sequencing. Next, we performed double sequential digestion on the OL plasmid: first 4 μg of plasmid was digested with SpeI-HF (NEB #R3133) for 1 h at 37 °C, heat inactivated at 80 °C for 20 min. Then, EagI-HF

(NEB #R3505) was added to the reaction and the mixture was incubated again at 37 °C for 1 h, and heat inactivated at 65 °C for 20 min.

We PCR-amplified the library using Q5 High-Fidelity DNA polymerase (NEB #M0491), 10 ng OL template in each reaction, 0.5 µl of each primer (in concentration of 100 µM), 1 µl dNTPs (10 mM). Please see Supplementary Data 1 for sURS OL amplification primer sequences. In total, we amplified in 40 50 µl reactions, to conserve the pool's complexity, which resulted in DNA concentration of 60.8 ng/µl (overall 60 µl), measured by Qubit. We set the PCR conditions using the manufacturer's protocol for 14 cycles, with annealing temperature of 65 °C, and 30 s for elongation time. Next, we performed two sequential digestions on the OL: first, 150 ng of OL were digested with 0.5 µl SpeI-HF for 1 h at 37 °C, heat inactivated at 80 °C for 20 min. Second, 0.5 µl EagI-HF was added to the reaction and the mixture was incubated again at 37 °C for 1 h, and heat inactivated at 65 °C for 20 min.

To clone the OL into the plasmid, both the plasmid and the OL amplicon pool were first digested with SpeI-HF and EagI-HF in multiple restriction reactions. The reaction products were then ligated together with T4 DNA ligase at 22 °C for 2 h, then heat-inactivated at 65 °C for 20 min. To conserve the OL complexity, we performed 40 identical ligation reactions with a plasmid:insert molar ratio of 1:1. Generally, 150 ng plasmid (5442 bp) and 3.528 ng OL (126 bp) were ligated in a 20 µl ligation reaction. We purified all OL amplifications and other enzymatic reactions using Promega Wizard DNA Clean-Up System according to manufacturer's protocol (Promega #A7280).

### Bacterial transformation and OL plasmid purification
We used 25 µl *E. cloni* 10G CLASSIC Electrocompetent Cells (Lucigen #60117 to transform 2 µl from the ligated OL-plasmid product. In total, 15 transformations were done. We electroporated cells using constant settings (1600 V for 5 ms). Immediately after electroporation, cells were recovered with 975 µl recovery medium (supplemented by Lucigen) for 1 h in 37 °C, 250 rpm. After recovery, batches of 250 µl cells were plated on LB-agar+ampicillin 140 × 20 mm petri dishes and grown overnight at 37 °C. For quality control and colony counting, recovered cells from each electroporation were plated on 2 LB+ampicillin 90 × 15 mm petri dishes: first, 20 µl recovered cells were diluted in 180 µl LB. Then, 10 µl and 50 µl diluted cells were plated. We used the number of colonies on these plates to estimate the overall number of transformants in a single electroporation. We estimated the total number of transformant to be $5.6 \times 10^6$ cells.

In addition, to preserve the OL as glycerol stocks, we cultured 100 µl recovered cells in 200 ml LB+ ampicillin overnight at 37 °C in 250 rpm shaker. The next day, cells were pelleted in 4 °C and re-suspended in ~6 ml liquid LB. Cells were divided into 1 ml aliquots and frozen immediately with 600 µl sterile 80% glycerol:20% water in −80 °C freezer.

We extracted OL plasmids from the plated bacteria. First, cells were scraped from the plates with 10 ml LB per plate. The resulting ~600 ml mixture was centrifuged (10 min, 4000 × g at 4 °C) and the plasmids were purified from the bacterial cells using Qiagen's Plasmid Maxi kit (#12162) following the manufacturer's manual. We cleaned each volume of 150 ml mixture separately using one column from the kit. For the final elution, 400 µl of TE buffer were used. We combined all maxi prep eluates into one tube, and concentration (~1800 ng/µl) and purity were measured by Nanodrop.

### Yeast integrations
We linearized purified OL plasmid by ApaI (NEB #R0114) restriction in the *URA3* gene in the plasmid for yeast genomic integration at the *URA3* locus. Overall, 40 digestion reactions were performed, where each had 8 µg plasmid and 1 µl ApaI in a total of 20 µl reaction. Reactions were incubated for 2 h at 25 °C and heat-inactivated at 65 °C for 20 min.

We cultured yeast W303-1A strain (W303-1A MATalpha leu2 his3 ade2 trpl ura3 was a gift from the Yoav Arava lab, Department of Biology, Technion - Israel Institute of Technology, Haifa, Israel) in liquid YPD medium (2% Glucose, 2% Bacto Peptone, 1% Yeast extract) at 30 °C and 250 rpm overnight. The next morning, culture's OD600 was determined by spectrophotometer, and the yeast cells were diluted to OD600 = 0.3 in 350 ml YPD media, and cultured in seven batches of 50 ml, each in a 500 ml flask. Cells grew until OD600 reached ~1, and then 50 ml cultured cells were harvested by centrifugation for 4 min at 3200 × g and washed with 20 ml sterilized deionized water. We added 1 ml of 0.1 M lithium acetate (LiAc). Cells were centrifuged for 3 min at 2400 × g, and pellet was re-suspended in 0.1 M LiAc. Subsequently, for each 100 µl cells, the cells were centrifuged again and 24 µl sterilized water, 36 µl 1 M LiAc, 5 µl of boiled single-stranded DNA (10 mg/ml salmon sperm ssDNA, Sigma-Aldrich #D7656), and 20 µl of ApaI-digested OL vector and 240 µl 50% polyethylene glycol were added. We incubated cells at 30 °C for 30 min and then transferred them to 42 °C for 15 min. After the heat shock at 42 °C, cells were spun down for 3 min at 2400 × g and resuspended in 200 ml SD with 2% glucose, supplemented with tryptophan, leucine, and histidine amino acids, divided into four batches of 50 ml, each in a 500 ml flask. Lastly, we grew cells at 30 °C for 60 h at 30 °C, 250 rpm. After two days of growth, we took 15 ml from each 50 ml culture for glycerol stock. Cells were spun down for 4 min, 3200 × g and resuspended in 1.5 ml sterile 50% glycerol:50% water.

### SORT-seq and OL genomic amplification
We grew yeast cells with integrated OL in liquid media (SD + 2% glucose with supplement of amino acids, without uracil) for 60 h, at 30 °C and 250 rpm. At the third day, cultures' OD were measured, and 10–15 ml batches of the grown cells were centrifuged (4 min, 3200 × g) and resuspended in 50 ml fresh SD + 2% glucose, supplemented with tryptophan, leucine, and histidine amino acids. A day later, 10 ml cultured cells were spun down and washed with 20 ml sterile deionized water. Cells were resuspended in fresh 50 ml PBSF (50 ml PBSX1 supplemented with 50 µl BSA X100, 20 mg/ml, NEB #B9200S). We kept resuspended cells on ice until the FACS run. As controls, WT W303-1A, weak mCore1 promoter, and strong FEC-mCore1 promoter, were analyzed as well. Integrated yeast cells were analyzed and sorted into 4 yeCitrine fluorescence bins using a BD FACSAria-IIIu cell sorter. Bin numbers represent sequential ranges of fluorescence, with bin1 for cells with the lowest fluorescence and bin4 for cells with the highest fluorescence. In each bin, at least 10 million cells were collected and kept on ice until the subsequent step. Next, we harvested binned cells in 4 °C for 10 min at 4000 × g. Supernatant was discarded carefully by pipet aid, leaving ~1 ml of PBSF (to avoid suction of cells). Binned cells were re-grown overnight at 30 °C, 250 rpm in 25 ml rich media (SD + 2% glucose) and supplemented with tryptophan, leucine, and histidine amino acids. The following morning, OD absorbance (expected 4–10 OD) was determined, binned cells were centrifuged for 4 min at 4000 × g, and the supernatant was discarded carefully. Next, for each of the four binned cultures, we isolated genomic DNA from the cells in the "Bust n' Grab" method[34] as follows: for each 1.5 ml of centrifuged medium, 200 µl of lysis buffer (1% SDS, 2% Triton X-100, 100 mM NaCl, 10 mM Tris-HCl pH = 8, and 1 mM EDTA pH = 8) was added to resuspend the cells. Cells were frozen in −80 °C for at least 2 min, to ensure complete freeze. Then, the cells were immediately heated in a 95 °C for 1 min to quickly thaw, following vigorous vortex for 30 seconds. 200 µl chloroform was added to each tube, followed by another vigorous vortex for 2 min. Tubes were centrifuged at 17,000 × g for 6 min at room temperature. The aqueous phase (containing the genomic DNA) was transferred to a tube with 400 µl ice-cold 100% ethanol. DNA was precipitated for 5 min at room temperature and then centrifuged 10 min at 17,000 × g at room temperature. Supernatant was removed and DNA pellet was washed with 500 µl 70% ethanol (diluted with ultra-

pure water, UPW). Tubes were centrifuged at $17,000 \times g$ for another 10 min at room temperature and supernatant was discarded. Ethanol was evaporated by heating the tubes for 5 min at 60 °C. DNA was resuspended in 50 µl TE (Biolab, 10 mM Tris, 1 mM EDTA pH = 8) or UPW and incubated at 65 °C for 10 min. We read samples' genomic DNA concentration using Nanodrop and stored at −20 °C. The OL genomic regions for each bin were amplified using QuantStudio Real-Time PCR System (Thermo Fisher) for 23 cycles in a 50 µl reaction with additions of: 1 µg genomic DNA, 0.5 µl Q5 High-Fidelity DNA polymerase, 1.5 µl 100% DMSO, 1 µl 10 mM dNTPs, 10 µl 5X Q5 reaction buffer, 2.5 µl 20X Evagreen dye (Biotium #31000), and 2.5 µl 10 mM from each forward and reverse primers. Overall, we ran nine 50 µl PCR reactions. PCR reactions were cleaned using the Wizard® SV Gel and PCR Clean-Up system kit (Promega #A7280) following the manufacturer's manual. DNA concentrations were determined by Qubit Fluorometer using the dsDNA HS Assay kit (Invitrogen™ #Q32851). In addition, the pool's length and samples' quality control were assessed by TapeStation (Agilent). Finally, we sent the OL to three NGS runs: two Nova-seq and one Next-seq run, which generated a total of 1500 M paired-end reads.

### NGS data processing and read normalization

First, overlapped paired-end NGS reads were assembled using PEAR code version 0.9.8[35]. Second, for ease of use and to fit into memory, the fastq files were split into 1M-read files using the SeqKit tool[36]. All split files were subsequently analyzed in MATLAB. The reads were counted according to their specific bin barcodes and variant barcodes, as follows: first, reads with lower base count (<200 bases) were filtered out from the analysis. Second, the 5' primer used to amplify the OL was identified at the 5' end of the variant's read. Third, the bin barcode in each read was verified, and then categorized for its respective bin. Note, the barcodes were designed with a Hamming distance of three enabling us to include barcodes that were not fully annotated with up to two misread or unannotated bases (i.e., N's). Next, for each identified bin barcode, we screened the variant barcode in the read and compared it to the OL design. Finally, we further analyzed reads with corresponding barcodes for their match with the full sURS variant sequence in the design. For the variants containing K/M mixed bases, a further analysis step ensued. Here, K/M bases were first noted according to their positions in the variant's design. Next, for each K/M position in the design, G or T (K base) and A or C (M base) were annotated. This allowed us to quantify the number of sub-motifs that appeared in the reads for each K/M containing-motif in the design. For every variation of K's and/or M's in the variant, the mean fluorescent expression level was calculated as well.

In order to assess the mean fluorescent expression level (Expression (A.U.)) for mixed-based motifs, non-mixed-base motifs, and sub-motifs, we normalized reads in each bin as follows: (1) For each bin, we calculated the total number of reads. (2) For each variant in a specific bin, we divided its read count by the total number of reads of that bin. (3) We calculated the percentage area of each bin ("%Bin") in the histogram derived from the sorting experiment. (4) We multiplied the outcome in step 2 by the corresponding "%Bin", resulting in adjusted reads per variant per bin. (5) For each variant, we summed the adjusted reads over the four bins. (6) Finally, for each variant, we divided the adjusted reads per bin from step 4 by the sum calculated in step 5, resulting in normalized reads for each bin.

After normalizing the reads, we calculated the mean fluorescent expression level for each variant in the library as follows: Expression (A.U.) = MeanBin1*x + MeanBin2*y + MeanBin3*z + MeanBin4*w, where: MeanBin1 = 607, MeanBin2 = 1364, MeanBin3 = 2596, and MeanBin4 = 7541. MeanBini are the mean fluorescent expression level that were measured for bins i = 1, 2, 3, 4 during the cell sorting, and x, y, z, and w are the normalized read counts in each bin for the specific variant.

### Minimum-hypergeometric analysis

mHG is a statistical analysis methodology with a specific application as a computational method to discover motifs in ranked lists of DNA sequences[25]. The mHG procedure notes whether a motif is present in each of the sequences and labels these as '1', and sequences without the motif are labeled as '0'. In the case of motif enrichment at the top (or bottom) of the list, the list is divided into 2 subsets: (1) the target set: the part of the list in which the motif is highly enriched, and (2) the background set: the remainder of the list, with lower occurrences of the motif. For each motif, the mHG determines the optimal partition cutoff in the list to separate between enriched and non-enriched sequences. It calculates the mHG *p*-value of the motif enrichment based on hypergeometric probability of the order within the ranked list. In addition, several other outputs are available: the motif's IUPAC representation, its PWM, and the optimal partition cutoff.

Herein, we used a python code implementation of mHG[25,37] to calculate the mHG *p*-value for each of the 41 motifs and the desert motif individually both for the activating and repressing effects. For each motif, we sorted all 2435 variants from highest to lowest mean fluorescent expression level (to find activating enriched motifs), and from lowest to highest (for repressing enriched motifs).

For the sub-motif analysis, we first "expanded" the 2098 variants containing motifs with mixed bases (of the 2435 variants with 22 barcodes each) into all possible variant sub-motifs (containing A/T/G/C in place of mixed bases). This resulted in the original 21 non-mixed-base motifs and an additional 80 sub-motifs derived from the mixed-base motifs. Afterwards, we applied the mHG analysis over each sub-motif. We present mean fluorescence expression levels for each of the sub-motifs in Fig. 4b–e and Supplementary Fig. 4.

### Machine-learning model to predict mean fluorescent expression levels

We developed a CNN to predict the mean fluorescent expression level (Fig. 3a). The input to the network is a one-hot encoded DNA sequence. We tested two inputs: the 101 nt-long variant sequence, or the 116 nt-long variant and barcode sequence. The input goes through multiple 1D-convolutional kernels, a global max-pooling layer, a fully connected layer, and finally an output layer. The output layer is either a single neuron with linear activation or four neurons with softmax activation, depending on whether we aimed to predict the mean fluorescent expression level or the distribution of normalized read counts across the four bins. We trained the network with MSE loss function for mean fluorescent expression level prediction and an average-weighted cross-entropy loss function to predict the normalized bins distribution, ADAM optimizer, and with or without sample weights which we defined as the total read count for each variant. The hyper-parameters were: 1024 kernels of width 6 with ReLU activation, 16 neurons with ReLU activation, 5 training epochs, and batch size 32. We did not test other hyper-parameters as results were already good with our initial choice. We developed the network using Keras and TensorFlow python packages.

We finally selected the architecture of the ADM, AMM, and MBO models with the following attributes: predicting a mean FL value for each variant, utilizing sample weights during the training process, and omitting the 15 nt-long barcode region from the input sequence. To improve prediction performance and robustness of the AMM and MBO models, we used a random ensemble initialization technique[38]. We trained 100 models on the datasets differing in the initial weights and training batches and used the average over their predictions as the final prediction. To further improve prediction performance and enable prediction over sequences longer than those in the training set, we added a 41-long motif count vector as input that is concatenated following the max-pooling layer, and uniform-padded the training input to the maximum prediction length.

## Validation experiments

Two sets of variants (see Supplementary Data 6 for sequences) were designed and synthesized by IDT as gBlocks gene fragments. The first set of variants includes 11 variants from the sURS OL, differing by their FL MBO-predicted output (2700–3400), chosen after analysis of the NGS and ML model development. The second set of variants consists of 32 variants, based on the significant motifs found via the mHG analysis. For this set, we designed a longer desert sequence in silico (186 bp in length) to test longer variants than those in the original OL (101 bp).

For validation in yeast cells, all variants were cloned individually upstream to the weak mCore1 promoter in the same manner that the sURS OL was cloned. Vectors were transformed into bacterial *E. coli* TOP10 competent cells, and positive colonies were verified after colony PCR by Sanger sequencing. Verified plasmids were purified from bacteria and digested with ApaI, for subsequent integration to the yeast genome in *URA3* locus as described previously. After integration, 2–8 colonies were picked and tested as single variants using flow cytometry. Colonies were grown in liquid SD medium at 30 °C with either 2% glucose, 2% glycerol, 2% glucose + 1NaCL, or at 39 °C with 2% glucose. All growth media was supplemented with leucine, histidine, and tryptophan amino acids. After overnight growth, cells were diluted and grown for 12 h, until reaching OD600 of 0.7–2. Then, cells were washed and resuspended in PBSF. Lastly, cells were analyzed in the MACSQuant VYB flow cytometer (Miltenyi Biotec).

In addition, part of the second set of variants (see Supplementary Data 6 for sequences and used motifs) was cloned upstream to the strong FEC-mCore1 promoter. The variants that were used for this validation contain either repressing motifs to down-regulate the promoter activity, or strong activating motifs to enhance the high activity of the promoter even more.

For the validation in a mammalian model CHO/K1 (CHO-K1/M1H strain was a gift from Oshimura lab, Graduate School of Medical Science, Tottori University, Tottori, Japan) and HeLa cells (HeLa strain was a gift from Esther Meyron-Holtz lab, Department of Biotechnology and Food Engineering, Technion – Israel Institute of Technology, Haifa, Israel), the validation variants were cloned upstream to a minimal CMV promoter and the *mCherry* target gene in pTRETightBI-RY-0 vector after removal of the TetO operator. pTRETightBI-RY-0 was a gift from Phil Sharp (Addgene #31463). The TetO operator was removed by plasmid digestion with XhoI and ApaI restriction enzymes, and the backbone was cleaned from gel using the Wizard DNA clean-Up System kit (Promega). Two complementary oligos with phosphorylated XhoI and ApaI ends encoding the minimal CMV promoter were annealed (see Supplementary Data 1 for sequences). The minimal promoter sequence was taken from the pTRE-Tight-BI-AcGFP1 vector from Clonetech (#631066). The annealed product was ligated using T4 ligase to the XhoI- and ApaI-restricted backbone. Afterwards, to clone the variants upstream the pCMVmin promoter, the backbone was cut with XhoI and KpnI-HF, cleaned by Promega Wizard kit and ligated with XhoI and KpnI-digested validation variants. The ligated plasmids were transformed to *E. coli* TOP10 cells, and positive clones were verified by sequencing. A positive control with a full CMV promoter was created by amplifying the promoter from the commercially available pTwist-CMV-BetaGlobin plasmid, also adding XhoI and KpnI sites at the ends of the promoter. The amplified promoter was digested with XhoI and KpnI-HF and ligated with the backbone, upstream to the *mCherry* gene. In addition, the CMV enhancer region was cloned upstream to the pCMVmin promoter, as a control to restore the promoter's activity in the CHO cells. All verification plasmids were Sanger sequenced.

## Mammalian cell experiments

For both maintenance and experiments, cells were grown in the same media. CHO/K1 cells were grown in F12 medium (Sartorius Cat. 01-095-1A) and HeLa cells were grown in DMEM High Glucose medium (Sigma Cat. D5796). Both media were supplemented with 10% FBS (Sartorius Cat. 04-007-1A) and 1% penicillin/streptomycin (Biowest Cat. L0022) solutions. Incubation conditions for growth or experiments were 37 °C and 5% $CO_2$. Cells were seeded into 96-well plates at a seeding density of $1.2 \times 10^4$ CHO cells or $2.4 \times 10^4$ HeLa cells in a 100 µl of media per well. 24 h post-seeding, media was changed into 100 µl fresh media. Per reaction, 0.65 µl of linear PEI (Mw 25 kDA, PolySciences Cat. 23966) in a final volume of 2.5 µl OptiMEM buffer (Thermofisher Cat. 31985070) was added to 100 ng of plasmid DNA in a final volume of 5 µl OptiMEM buffer, altogether making up 7.5 µl of reaction per well. The yielded a DNA-PEI mixture was left to incubate for 15 min before being added to each well. 24 h post-transfection, media was changed again into 100 µl fresh media, removing media containing PEI. 72 h post-transfection. Media was removed, and cells were washed once with 50 µl of 1xPBS (Sartorius Cat. 02-023-1A) per well. PBS was removed and 25 µl of 0.05% trypsin solution (Sartorius Cat. 03-054-1A) per well was added and the plate was incubated for 3 or 5 min (for CHO or HeLa cells, respectively) at 37 °C and 5% $CO_2$. Afterward, cells were resuspended using 100 µl of fresh media and measured using flowcytometry (MacsQuant). Lasers ($\lambda = 405$ nm and 560 nm) parameters were 224 V, 235 V, 200 V, and 208 V for FSC, SSC, Y2, and V1 lasers, respectively, with an FSC trigger of 8.4.

## Statistics and reproducibility

Oligo library was sequenced three times. First, directly by amplifying the purchased Twist DNA variants. Second, after cloning into plasmids via *E.cloni*. Third, after integration into yeast cells and sorting via flow cytometry. Yeast Integrated oligo library experimental sorting and extraction of promoter library was carried out once. Libraries for NGS sequencing amplified from extracted yeast integrated promoters were assembled twice on separate dates and sequenced separately.

Each validation variant characterization experiment was carried out via flow cytometry, repeated in technical duplicates and carried out in biological triplicates over three days. This was done for all 43 validation variants in yeast, and the 32 validation variants in CHO and HeLa cells.

No statistical method was used to predetermine sample size. No data were excluded from the analyses. The experiments were not randomized. The Investigators were not blinded to allocation during experiments and outcome assessment.

## Reporting summary

Further information on research design is available in the Nature Portfolio Reporting Summary linked to this article.

## Data availability

The raw sequencing files generated in this study are available on the SRA website under BioProject accession number PRJNA1061345, Bio-Sample accessions: SAMN39265674, SAMN39265675, SAMN39265676, SAMN39265677, SAMN39265678, SAMN39265679, SAMN39265680, SAMN39265681. Source data are provided with this paper.

## Code availability

The code, trained models, and processed datasets are publicly available via github.com/OrensteinLab/UniLib[39].

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

## Acknowledgements

This work was greatly supported by the European Union's Horizon 2020 Research and Innovation Programme under grant agreement no. 851615. This study was partially supported by Israel Innovation Authority. We thank Orna Atar, Noa Katz, Michal Meirom-Brunwasser, Naor Granik, Bea Kaufmann, Roni Cohen, and Noa Eden-Navon for their invaluable input during the study. We thank Hadas Yung and Tomer Antman for their help with the OLs' cloning and transformations. We thank Roy Shafir, Leon Anavy, and Zohar Yakhini for their help in the UNILIB OL design and beneficial input on the NGS data.

## Author contributions

R.A. and I.V. envisioned and devised the idea of the study. I.V. designed and processed the UNILIB OL, cloned the OL in bacteria and yeast, performed the protocols, yeast experiments and analysis, designed and performed the validation experiments, and wrote the manuscript. S.G. designed part of variants in the validation experiments. O.W., Y.Z. and S.G. constructed part of the validation variants and performed the validation in CHO and HeLa cells. Y.O., J.M., D.B.A. and H.H. developed and tested the ML models. Y.O. and S.G. provided comments during the study. R.A. performed part of the analysis, supervised the work, and wrote the manuscript.

## Competing interests

The authors declare no competing interests.
