## [Peer Review File · Nature Communications]

Reviewers' Comments:

Reviewer #1:

Remarks to the Author:

In this study, the authors introduce an algorithm designed to predict regulatory sequences that modulate gene expression in eukaryotic cells. To achieve this, they identified 41 Transcription Factor Binding Sites (TFBS) from various organisms, including *S. cerevisiae*, *S. pombe*, *D. melanogaster* S2 cells, and murine ES cells. These TFBS (or motifs) were used to construct a library of nearly 190,000 variants, each containing up to three different motifs and their variants. Using a combination of machine learning and oligo library analysis in yeast, the authors aimed to identify specific motifs capable of enhancing or attenuating gene expression. These motifs were then assessed in two mammalian cell lines.

Overall, the study addresses a topic of general interest: fine tunability of gene expression for optimal output production. However, the manuscript suffers from issues related to clarity, methodology transparency, and validation, ultimately raising concerns about the rigour and robustness of the study. Detailed comments and suggestions are provided below.

1. The TFBS used in this study are for endogenous TFs, and some correlate with both activation and repression activity. This may depend on different cell types, cell states and growth conditions, potentially generating high variability in the promoter's behavior. The authors should validate the identified sequences with a significant number of biological replicates (at least 3), at different cell passages and possibly different culturing conditions (e.g. different cell densities, different media composition, etc).
2. The authors should ensure that the cells carrying a specific barcode indeed harbour the intended promoter variant, especially considering the complexity of the designed library. Thus, one significant gap in the methodology is the absence of verification steps after sorting, ideally conducted after the final PCR step just before Next-Generation Sequencing (NGS). This validation step could take various forms, even only Sanger sequencing of a select subset of variants (not only 1!). This additional verification would provide confidence in the accuracy of the cell content and the presence of the intended promoter variants.
3. Figure 2c should ideally show a relatively tight cluster of data points resembling a straight line with some expected noise and a few outliers, reflecting the inherent variability in biological systems and NGS data. However, the observed data points in this plot appear scattered, indicating a significant divergence from expected behavior. To mitigate this concern and validate the Sort-Seq values, it is advisable to conduct an extensive validation experiment. This experiment could involve the measurement of fluorescence or RNA-seq for individual variants, followed by a correlation analysis with the NGS-based fluorescence (FL mean) values. If this validation does not yield a strong correlation, it could raise doubts about the reliability of the entire experimental approach and data interpretation. Therefore, addressing these issues through thorough verification and validation steps is crucial to enhance the study's credibility and confidence in the results.
4. Increased gene expression, can correlate with reduced host organism fitness (DOI: <https://doi.org/10.1038/nmeth.3339>) and unwanted competition for intracellular resources (DOI: <https://doi.org/10.1038/s41467-020-18392-x>). If the final goal is to characterize new promoters for bioproduction, the authors should show that they do not impact the fitness of the host cells.
5. Both introduction and discussion need more supporting citations. Please add evidences to support all claims. Additionally, some claims seem only partially true or imprecise:
 - a. Line 47: there are studies that addressed the same problem (e.g. DOI: <https://doi.org/10.1128/aem.00939-22>). These should be cited and discussed.
 - b. Lines 81-85: these claims should be either supported by citations or toned down. There are many well characterized constitutive promoters active across eukaryotic species (CMV, EF1a, PGK, etc). The authors should motivate why their design is superior to existing solutions.
6. Some experimental choices and pipeline steps are unclear:
 - a. How were the 41 TFBS selected, especially since many are not common across the 4 eukaryotic cell types evaluated?
 - b. Line 128: how was the threshold of >73% determined and why?
 - c. In the method section, describe how the total of 1600 million NGS reads mentioned is processed and the steps involved. Why only 25% of NGS reads were selected? What is the potential impact on the results?
7. Some of the conclusions drawn by the authors in the results section are questionable. Can the

authors comment on the following points?

- a. In Fig. 2a the authors claim that there is a clear correlation between the median of the mean fluorescence and the motif: this is not clear from this graphic and it could be pure randomness.
 - b. Line 236: the observed correlation indicates that the distribution is mainly not described by the model.
 - c. Line 269-271 and Fig. 3d: there is no clear correlation between an increase in mixed letters and increase in model performance. The values of 10, 26, 36, 16, 23, 18% of correlation could be more or less random numbers.
8. The manuscript presentation is unclear in some parts, especially in the results and discussion sections preventing full understanding of the study:
- a. The term "mean FL" is a function of the NGS reads. This is confusing as "mean FL" should intuitively be the mean of the measured fluorescence only.
 - b. Is "mean fluorescence" (Fig. 2a) same as "mean FL" (Fig. 2c and others)?
 - c. Large parts of the results section suffer from a lack of clarity, for example lines 168-190. Please work on improving the clarity of the entire section.
 - d. The structure of the discussion is confusing: start with a brief summary of the work, then draw a conclusion and hypothesize the next steps, only to return to the summary in the middle of the second paragraph. Consider restructuring.
9. There are several overstatements in the manuscript that should be toned down. E.g.:
- a. Line 106 "pan-organism": the study focuses on one yeast strain and two mammalian cell lines only.
 - b. Line 112 "different murine tissues": this is inconsistent with the caption of Supplementary Figure 1 where only mouse embryonic stem cells are mentioned.
10. Some scientific basics are lacking:
- a. space between number and units
 - b. organism in italic
 - c. missing descriptions in the methods (see comment 6c)
 - d. missing code and data online: will these be provided after publication?
 - e. the unit of fluorescence is never stated, is it arbitrary units (au)?
 - f. number of biological replicates should be specified. Especially Figure 6 does not have error bars: was validation run on one biological replicate only? If this is the case, it is scientifically inaccurate to draw any conclusion.

Minor comments:

1. Line 49: "whose promoter is capable of transcribing" is wrong. A promoter does not transcribe, the polymerase does.
2. Line 102: "reliable prediction"
3. Line 106: "constitutive" instead of non-inducible
4. Line 156: "Fig. 1f"
5. Line 802: what is a PPM?
6. Line 140: Fig. 1c?
7. Line 185: which type of correlation is it?
8. Line 422: "for a weak and a strong promoter", only one variant tested for each
9. Line 430: "circuits" (Remove "bio- ")
10. Line 869: "of"s" ?!
11. Lines 780-781: lasers wavelength?
12. Discussion: The name UNILIB is introduced only at this point, why?
13. Figures:
 - a. Increase the font size in all figures, as they appear too small for easy readability
 - b. 1f: the vertical line should be at 40, it is at 30 though
 - c. 2d: no error bars
 - d. 3e: labels can be put under x axis instead of colors
 - e. 3d: if a linear correlation (Pearson) is assumed, add the line to the plot
 - f. 4a: error bars are at null level?
 - g. 4b (and following): no tick marks in the right plots
 - h. 5f: error bars
 - i. 6: add axis label, what type of data is shown?

Reviewer #2:

Remarks to the Author:

In this manuscript from Vaknin et al, the authors develop a library of synthetic promoters by merging a target core promoter and upstream TF binding sites, to enhance protein expression in eukaryotic cells. This is not the first time synthetic promoters are designed by this approach and previous work has shown that is possible to engineer mammalian cells based on mining of TFBS and core promoter engineering (e.g. Johari et al. 2019 and others).

The manuscript tries to go beyond the state of art by building a system functional in more than one chassis.

I have a few questions, mainly on the experimental side, as I am not a computational person. It is not fully clear to me what the authors mean with the term "motif" and it could be useful to define this.

It is not fully clear also how the 41 motifs were identified and if this was validated by using more than one TF data base.

In figure 1e it is not fully clear why authors have not taken population 2 and re-bin it in order to capture more variants and potentially more interesting candidates with higher expression. It could have been beneficial to do so here.

In figure 6, for CHO cell experiments, the authors mentioned they used a BFP for fluorescence normalisation. However, it is known from literature that resource competition can impact normalisation (see Frei et al, 2020. Jones et al, 2020).

Can the authors shown the expression levels of the CHO library but with no normalisation and check if that improves the results? Caption of figure 6 should also more clearly describe what the figures shows.

Minor comments pertain to several typos present in the manuscript like Eukaryotic that should be lower case;

Figures are called at time with capital letter and at times with lower case.

In conclusion I suggest the work to be published once these questions have been addressed.

Reviewer #1 (Remarks to the Author):

In this study, the authors introduce an algorithm designed to predict regulatory sequences that modulate gene expression in eukaryotic cells. To achieve this, they identified 41 Transcription Factor Binding Sites (TFBS) from various organisms, including *S. cerevisiae*, *S. pombe*, *D. melanogaster* S2 cells, and murine ES cells. These TFBS (or motifs) were used to construct a library of nearly 190,000 variants, each containing up to three different motifs and their variants. Using a combination of machine learning and oligo library analysis in yeast, the authors aimed to identify specific motifs capable of enhancing or attenuating gene expression. These motifs were then assessed in two mammalian cell lines. Overall, the study addresses a topic of general interest: fine tunability of gene expression for optimal output production. However, the manuscript suffers from issues related to clarity, methodology transparency, and validation, ultimately raising concerns about the rigour and robustness of the study. Detailed comments and suggestions are provided below.

1. The TFBS used in this study are for endogenous TFs, and some correlate with both activation and repression activity. This may depend on different cell types, cell states and growth conditions, potentially generating high variability in the promoter's behavior. The authors should validate the identified sequences with a significant number of biological replicates (at least 3), at different cell passages and possibly different culturing conditions (e.g. different cell densities, different media composition, etc).

We thank the reviewer for this comment, and the opportunity to improve our work. First, we would like to note that only 27 of the 41 motifs were validated transcription-factor binding sites. The rest of the motifs were unknown, and not attributed to a particular regulatory process. The motifs were found to be conserved across various Eukaryotic lineages in a HT-SELEX assay carried out by Jussi Taipale's group (published in DOI: 10.1038/nbt.4138). As a result of the UNILIB experiment, we were able to characterize 5 of the unknown motifs as either up- or down-regulating. Some of these motifs are weak and require multiple sites or combinations of sites to show a measurable regulatory effect.

Second, with regards to the function of the motifs in different cell-growth conditions or cell types, there indeed could be a variation. Consequently, we created 43 unseen synthetic URS variants. We used 32 variants, which sampled 23 of the motifs for which a significant up or down-regulatory behavior was identified, for the results depicted in Fig. 5, and an additional 11 variants were used for the validation of the machine-learning model in Fig. 3. Altogether the 43 validation variants sample 31 of the 42 motifs including all the motifs that were found to be significant.

However, given the reviewer's critical comment, we decided to use these variants to expand our validation set to further strengthen our conclusions. The revised manuscript now includes two new figures: Fig. 6 and Fig. 7, which replace the original Fig. 6. New Fig. 6 details the results of 7 separate experimental measurements carried out in biological triplicates as follows:

1. Yeast: SD-Ura+2% glucose, 30°C, weak core promoter (mCore), no additives, 43 variants.
2. Yeast: SD-Ura+2% glycerol, 30°C, weak core promoter (mCore), no additives, 43 variants.
3. Yeast: SD-Ura+2% glucose, 39°C, weak core promoter (mCore), no additives, 43 variants.
4. Yeast: SD-Ura+2% glucose, 30°C, weak core promoter (mCore), 1M NaCl, 43 variants.
5. Yeast: SD-Ura+2% glucose, 30°C, strong core promoter (FEC-mCore), no additives, 20 variants.
6. CHO/K1 cells grown in F12 medium, 32 variants.
7. HeLa cells grown in DMEM high glucose medium, 32 variants.

The results for all the yeast experiments, whether as biological replicates or in different culturing conditions, correlate strongly with one another in a statistically significant fashion (see below Figs. L1 in addition to Fig. 6 and Supplemental Fig. 6 in the revised manuscript). These results support a conserved function in yeast independent of growth conditions. This is consistent with the previous findings of (Keren et al., Mol.

Sys. Biol., 9:701 (2013)) who tested the activities of ~900 *S. cerevisiae* promoters in 10 different conditions and found no significant difference in activities across the panel of growth conditions used. In addition, the experimental data also shows a statistically significant correlation between the mammalian cell measurements and the yeast measurements, which we showed also in the original manuscript. Finally, we also improved our modelling scheme by extending the machine-learning model developed in Fig. 3 using the additive model described in Fig. 5 to create a hybrid model, which improved performance on all datasets. The new hybrid model now predicts HeLa and CHO cells regulatory activity in a statistically significant fashion (see new Fig. 7).

The revised validation segment of our manuscript now allows us to make an improved argument that the synthetic URS sequences that we created can generate a regulatory function that is, for the most-part, independent of cell-type and growth conditions, making these sequences a potentially widely applicable tool.

Figure L1: Comparison of biological triplicates for the 43 validation variants grown in SD-Ura+2% glucose. Experiments show a strong correlation between all 3 repeats.

2. The authors should ensure that the cells carrying a specific barcode indeed harbour the intended promoter variant, especially considering the complexity of the designed library. Thus, one significant gap in the methodology is the absence of verification steps after sorting, ideally conducted after the final PCR step just before Next-Generation Sequencing (NGS). This validation step could take various forms, even only Sanger sequencing of a select subset of variants (not only 1!). This additional verification would provide confidence in the accuracy of the cell content and the presence of the intended promoter variants.

In brief, due to their short length (bc+sURS+core-promoter) our variants were fully sequenced in the NGS. While we apologize for not including this data in the manuscript, as part of our validation, we screened for all full-length reads in the library (i.e. reads that yielded a full length of over 200 bp that include the barcodes, sURS, and primer regions). We found approx. 400M such reads, and they all perfectly matched the barcodes, motifs, and primer regions. We then quantified the correctness of the downstream sequence to the barcode with the intended design and found that per position >99% match the design. In addition, we designed the barcodes, such that the Hamming distance between each two is at least 3, which maximized our ability to identify variants. We thank the reviewer for this important comment. In response to the comment, we have added a comment under the Methods sub-section “NGS data processing and read normalization” to clarify this issue.

3. Figure 2c should ideally show a relatively tight cluster of data points resembling a straight line with some expected noise and a few outliers, reflecting the inherent variability in biological systems and NGS data.

However, the observed data points in this plot appear scattered, indicating a significant divergence from expected behavior. To mitigate this concern and validate the Sort-Seq values, it is advisable to conduct an extensive validation experiment. This experiment could involve the measurement of fluorescence or RNA-seq for individual variants, followed by a correlation analysis with the NGS-based fluorescence (FL mean) values. If this validation does not yield a strong correlation, it could raise doubts about the reliability of the entire experimental approach and data interpretation. Therefore, addressing these issues through thorough verification and validation steps is crucial to enhance the study's credibility and confidence in the results.

We are aware of the lack of correlation observed in our OL measurements, which we view as a feature of the methodology rather than an artefact. The data used to compose panel 2c consisted of OL variants that were encoded with two barcodes. The lack of correlation observed for most variant pairs was due to three main reasons:

1. In SORT-seq experiments, it is impossible to tell how many reads came from a single sorted cell. For example, one cell can yield several hundred reads or a single read in the final count. This can lead to a lack of correlation if the number of cells sorted per a particular variant is small as in this case. We remind the reviewer that our library was unprecedented in size (~200K mixed-base variants), and thus the number of cells collected for each variant was small.
2. In SORT-seq experiments, it is impossible to tell whether cells die after sorting or divide. In such a case, a single cell may also yield more read events in comparison to another cell, though the original sort had them at an equal weight.
3. Variants which do not encode a particular regulatory function are more susceptible to transcriptional variation. The strongly up- and down-regulating variants (top and bottom corners of the plot respectively) are strongly correlated as expected from a fully functional regulatory element.

The analysis shown in Figure 2 and referred to by the reviewer was, therefore, used as impetus for the quantitative analysis that was done on the 2,435-variant subset that were encoded with 22 barcodes. We reasoned that the sources for the lack of correlation discussed in the points above will be minimized by analyzing multiple barcodes on the same variants. For example, identifying all 22 barcodes immediately implies that at the very least we screened 22 separate fluorescent yeast cells. Consequently, we reasoned that such a subset will lead to a more robust statistical analysis, even though a given pair of barcodes may not be as correlated as we would like.

The analysis carried out in Figure 3 shows that the choice of unifying the 22 barcodes is statistically robust as evident by the agreement with the de Boer machine-learning model that was trained using independent experimental data. In addition, the validation set of Figures 5-7 and the agreement with the models that were derived from the experimental data obtained from the 2,435-variant subset essentially shows that both the SORT-seq experiment and the analysis were robust. Consequently, we believe that we have provided sufficient proof in the manuscript both computationally and experimentally that our analysis and experimental approach were robust, and no additional validation is necessary.

4. Increased gene expression, can correlate with reduced host organism fitness (DOI: <https://doi.org/10.1038/nmeth.3339>) and unwanted competition for intracellular resources (DOI: <https://doi.org/10.1038/s41467-020-18392-x>). If the final goal is to characterize new promoters for bioproduction, the authors should show that they do not impact the fitness of the host cells.

We thank the reviewer for this comment. In response to the comment, we carried out fitness analysis on the 32 validation variants. The results are shown in Figure L2 and have been added to the Supplementary Information as Supplementary Fig. 6. In our experiments, we did not detect any effect on fitness as a result of our synthetic upstream regulatory sequences and mentioned that in the “Boosted expression level translates from yeast and mammalian cells” Results subsection of the revised manuscript. Specifically, we grew the 43 validation variants in YPD and tracked their growth via OD measurement as function of time

(Fig. L2a - circles). For each strain we then fitted the OD measurements (Fig. L2a – blue line) with the following model for exponential growth:

$$OD_{600}(t) = C + \frac{L}{(1 + e^{-k(t-t_0)})}$$

Where C is background OD levels, L is the max OD, k is the growth rate, and t_0 corresponds to the lag time (i.e. time at which the culture reaches OD of $L/2$). Using this model, we extracted the growth rate for each strain and for both repeats. We plotted in Fig. L2b the different fitted growth rates (k) for both repeats that were measured for each variant. The results show that the growth rates for all experiments was found to be within a narrow range of ~ 0.4 - 1.3 (1/hr) without any significant correlation between duplicates. The lack of correlation between duplicates and narrow range of growth rates provides strong evidence that our variants do not affect the fitness of the yeast cells.

Figure L2: Fitness analysis for yeast cells expressing the 43-variant validation set grown in YPD . (a) The 43 validation variants were grown in YPD in duplicates and tracked for OD as a function of time (circles). For each variant the growth data are fitted (blue lines) by a classic growth curve (see Supplementary Information). (b) The rates of growth for each variant are plotted as a scatter plot pair for both repeats.

5. Both introduction and discussion need more supporting citations. Please add evidences to support all claims. Additionally, some claims seem only partially true or imprecise:

a. Line 47: there are studies that addressed the same problem (e.g.

DOI: <https://doi.org/10.1128/aem.00939-22>). These should be cited and discussed.

b. Lines 81-85: these claims should be either supported by citations or toned down. There are many well characterized constitutive promoters active across eukaryotic species (CMV, EF1a, PGK, etc). The authors should motivate why their design is superior to existing solutions.

We thank the reviewer for providing the additional reference, and for pointing out misstatements in the Introduction. In response to the comment, we added this and other references and rephrased the text in the Introduction and Discussion sections accordingly. Please see the marked up version of the revised manuscript for the details.

6. Some experimental choices and pipeline steps are unclear:

a. How were the 41 TFBS selected, especially since many are not common across the 4 eukaryotic cell types evaluated?

We apologize that these choices were unclear. In brief, the motifs were chosen as part of a past collaboration with Jussi Taipale's group. At the time, the Taipale group had a study (later published in DOI: 10.1038/nbt.4138) that was focused on developing a protein activity assay specifically for DNA-binding TFs in cell and tissue extracts. His team was able to identify strong and enriched TFBS motifs using this assay, including the 41 TFBSs that we selected for our research. Those and other motifs were discovered in numerous organisms, including two types of yeast, different tissues from mice, bacteria, and *Drosophila* S2 cells. Even though some of the motifs were not found to be common/conserved across these organisms, we nevertheless chose to characterize them due to similarities to well-characterized protein families in higher eukaryotes (e.g. the bHLH motif). For a detailed explanation as to how and why we chose these 41 motifs, please see the new Methods subsection titled "Motif selection and encoding into OL. Based on the study's data, we selected the 41 enriched motifs according to the following criteria:

1. Different motif types: 8 organism-shared motifs, 5 mice tissue-shared motifs, 14 unshared motifs (unique to an organism), and 14 unknown motifs (with unknown regulatory function).
2. Known/unknown regulatory function: 27 of the selected motifs have known regulatory function, and those were anticipated to be the control motifs. Characterizing the regulatory function of the 14 unknown motifs in yeast was one of the stated goals of the study.

b. Line 128: how was the threshold of >73% determined and why?

We thank the reviewer for this question and apologize for not including the rationale for our choice in the original manuscript. In response to the comment, we have added a new Methods subsection entitled "Motif selection and encoding into OL", where the rationale is now described. In brief, K and M substitutions were based on the percentage calculation of the respective G/T and A/C occurrences, in each position of the motif according to the PFM data given to us by the Taipale group (later published in DOI: 10.1038/nbt.4138). Positions within the motif, with dominant percentages, were replaced by either K or M in the final design. 70% threshold was set to determine the K/M substitution, but the actual calculated threshold was higher at 73%, as specified in the Results section.

c. In the method section, describe how the total of 1600 million NGS reads mentioned is processed and the steps involved. Why only 25% of NGS reads were selected? What is the potential impact on the results?

We thank the reviewer for this question. In response to the comment, we have expanded the Methods subsection titled "NGS data processing and read normalization", specifying the processing steps accordingly. Given the size of the UNILIB library, we opted to use Illumina's NovaSeq for its ability to generate as much as 1B reads per run. We used Illumina's S1 kit, and despite Illumina's reliability claims, indeed only 25% of the reads were found to be correct in both the NovaSeq runs that were made. Neither we nor the Genomics Center at the Weizmann Institute are certain about the underlying cause for this low fidelity, which may have been due to the choice of primers. Irrespective of that, the center offered to re-sequence the library, ultimately allowing us to extract 400M correct reads. This number of reads proved to be sufficient for the analysis in our study, which was based on the machine-learning modelling.

As discussed above, our analysis showed that the use of two barcodes per variant was insufficient for proper quantitative analysis, as such a low number introduced a large statistical uncertainty regarding the actual number of sorted cells. On the flip side, the same analysis showed that 22 barcodes per variant significantly reduced this uncertainty. In the former case, increasing the number of reads would not have affected the cell number uncertainty, while in the latter the 22 barcodes ensured that we have a sufficient amount of reads for every one of the variants of the 2,435-variant subset even with only 400M correct reads extracted in total. Consequently, the 25% extraction rate did not impact our conclusions or results of the downstream analysis.

7. Some of the conclusions drawn by the authors in the results section are questionable. Can the authors comment on the following points?

a. In Fig. 2a the authors claim that there is a clear correlation between the median of the mean fluorescence and the motif: this is not clear from this graphic and it could be pure randomness.

We thank the reviewer for this comment and apologize for not assessing the correlation in the original manuscript. In response to the comment, to assess the correlation between the median of mean fluorescent expression levels and motifs, we conducted a Wilcoxon rank-sum test by comparing the mean fluorescent expression levels of the group of variants containing each motif to the group of variants containing motifs ranked at least 5 motifs away. To correct for multiple tests, we applied the Benjamini-Hochberg procedure with an FDR threshold of 0.1. This new analysis revealed significant differences for 36 out of 42 motifs (p -value <0.05), where the 6 non-significant motifs are in motifs ordered 15-20 in decreasing order, which is expected since their values are distributed around the center of the mean-fluorescence distribution. We added this new statistical significance analysis to the Results section.

b. Line 236: the observed correlation indicates that the distribution is mainly not described by the model.

We thank the reviewer for this comment. In various biological problems, such as inference of protein-DNA binding preferences based on protein-binding-microarray data, correlation values are within 0.4-0.6 ([10.1109/TCBB.2019.2947461](https://doi.org/10.1109/TCBB.2019.2947461)). Moreover, the correlation of 0.45 was achieved by the all-data model (ADM), which we showed is inferior to the AMM and MBO models. In addition, this value is calculated on a test set of variants, where most have only 2 barcodes, which implies the lower quality of their mean-fluorescence measurements compared to variants with 22 barcodes. One of the key insights from our study was that variants supported by 22 barcodes, as opposed to 2 barcodes, yielded more accurate mean-fluorescence measurements (due to the inclusion of more cells carrying the variant). When the MBO model was exclusively trained and tested on datasets of variants with 22 barcodes, the obtained Pearson correlation was 0.61. This suggests that using more accurate training and test sets results in a higher observed correlation, which better explains the variability in the data.

c. Line 269-271 and Fig. 3d: there is no clear correlation between an increase in mixed letters and increase in model performance. The values of 10, 26, 36, 16, 23, 18% of correlation could be more or less random numbers.

We thank the reviewer for pointing this unclear part of our manuscript. We acknowledge that the reported correlation values may not achieve statistical significance. In response to this comment, we removed Fig. 3d and the corresponding text describing the analyses and results from the Results section.

8. The manuscript presentation is unclear in some parts, especially in the results and discussion sections preventing full understanding of the study:

a. The term “mean FL” is a function of the NGS reads. This is confusing as “mean FL” should intuitively be the mean of the measured fluorescence only.

We thank the reviewer for this comment and have changed all “mean FL” labels for the NGS data to the one used by (Sharon, E. et al. Nat Biotechnol 30, 521–530 (2012)) in a similar oligo library experiment: “Expression (A.U.)”. All relevant labels in the figures and text have been changed to Expression (A.U.) in the revised manuscript.

b. Is “mean fluorescence” (Fig. 2a) same as “mean FL” (Fig. 2c and others)?

We thank the reviewer for finding this minor error, and changed to Expression (A.U.).

c. Large parts of the results section suffer from a lack of clarity, for example lines 168-190. Please work on improving the clarity of the entire section.

We thank the reviewer for this comment and have clarified the Results, including the subsection titled “Variants manifest a broad range of regulatory behavior”. Please see the marked-up version of the revised manuscript for the changes.

d. The structure of the discussion is confusing: start with a brief summary of the work, then draw a conclusion and hypothesize the next steps, only to return to the summary in the middle of the second paragraph. Consider restructuring.

We thank the reviewer for this comment. We decided to leave the structure of the Discussion as is with minor modifications. We opted for this option so we can highlight at the beginning of the section the design algorithm that was developed, which we believe is the most important achievement of this work.

9. There are several overstatements in the manuscript that should be toned down. E.g.:

a. Line 106 "pan-organism": the study focuses on one yeast strain and two mammalian cell lines only.

In response to the comment, we deleted the term pan-organism from the Introduction.

b. Line 112 “different murine tissues”: this is inconsistent with the caption of Supplementary Figure 1 where only mouse embryonic stem cells are mentioned.

In response to the comment, we corrected the error and replaced the term “different murine tissues” by mouse (ES cells and different tissues).

10. Some scientific basics are lacking:

a. space between number and units

Errors were corrected.

b. organism in italic

Errors were corrected.

c. missing descriptions in the methods (see comment 6c)

All missing methods were added. Please see responses above.

d. missing code and data online: will these be provided after publication?

All data and code were uploaded. Please see data and code availability statements in the revised manuscript.

e. the unit of fluorescence is never stated, is it arbitrary units (au)?

The mean FL was replaced by the consensus Expression (A.U.) for all mean level expression level measurements.

f. number of biological replicates should be specified. Especially Figure 6 does not have error bars: was validation run on one biological replicate only? If this is the case, it is scientifically inaccurate to draw any conclusion.

We thank the reviewer for this comment and would like to clarify our choice of plotting. All validation experiments were conducted in (at least) biological triplicates as discussed above (see Fig. L1 for example). In response to Comment 1, we have substantially increased the volume of the validation experiments, and consequently split the original Fig. 6 in the original manuscript to a new Fig. 6 depicting only experimental data, and a new Figure 7 depicting the modelling analysis. Given the quantity of data, we opted to omit the error-bars from some of the plots for aesthetic reasons (see for example Fig. L3 which depicts Fig. 6a-c

with error bars). Since we recognize the need to visually and numerically report the variance in the validation measurements, we have done the following:

- The variation for the various validation datasets across the repeats can be assessed using new Supplementary Figure 6, and via the correlation heatmap plots of new Fig. 6 (panels d and e). Note, that the autocorrelation between the various repeats of the yeast experiments does not yield a higher Pearson correlation coefficient, as compared with the cross-correlation of the mean fluorescent expression levels of the various conditions. This result provides statistically significant experimental evidence that the different yeast growth conditions do not alter the expression of our variants, and in our opinion is depicted more convincingly via the mode of plotting that we chose to use in the revised manuscript.
- We plotted Fig.6a with error bars to convey directly the variance in the cross-correlation analysis. This panel is identical to the one shown in Fig. L3a.
- Fig. 6b-c are depicted without error-bars. Here, the purpose is to highlight the fact that out of the dataset 24 variants behaved similarly to yeast (blue), while 8 were uncorrelated in CHO (red). We have added the panel b and c to supplementary fig.6 as panel g and h.
- We plotted the error-bars for the both the yeast-glucose 2% and HeLa-blue variants in panel b and c of Fig. 7.

Figure L3: Cross correlation data for 43-variant validation experiments plotted with error-bars. (a) yeast cross-correlation data. (b-c) The 2% glucose yeast expression data plotted as a function of CHO (b) and HeLa (c) expression data. Error-bars were computed using standard-error analysis carried out on mean flow cytometry fluorescence measurements obtained from three or four biological repeats – depending on data set.

Minor comments:

1. Line 49: “whose promoter is capable of transcribing” is wrong. A promoter does not transcribe, the polymerase does.

We changed the statement to “whose promoter is capable of initiating transcription”

2. Line 102: “reliable prediction”

We removed the word prediction.

3. Line 106: “constitutive” instead of non-inducible

We retrained the word non-inducible.

4. Line 156: “Fig. 1f”

We corrected the typo.

5. Line 802: what is a PPM?

We corrected PPM to PWM – position weighted matrix.

6. Line 140: Fig. 1c?

We corrected the typo.

7. Line 185: which type of correlation is it?

We added the word “Pearson”.

8. Line 422: “for a weak and a strong promoter”, only one variant tested for each

We changed the phrase in accordance with the reviewer’s suggestion.

9. Line 430: “circuits” (Remove “bio- “)

We removed the word bio.

10. Line 869: “of”’s” ?!

We corrected the typo.

11. Lines 780-781: lasers wavelength?

We added MacsQuant laser wavelength.

12. Discussion: The name UNILIB is introduced only at this point, why?

We chose to name the completed design algorithm UNILIB, and name it only after we demonstrated that it worked via the validation experiments.

13. Figures:

a. Increase the font size in all figures, as they appear too small for easy readability

We corrected where possible.

b. 1f: the vertical line should be at 40, it is at 30 though

We corrected this error.

c. 2d: no error bars

There are no error bars for the medians of the mean fluorescent expression level distributions.

d. 3e: labels can be put under x axis instead of colors

We improved the figure accordingly.

e. 3d: if a linear correlation (Pearson) is assumed, add the line to the plot

We removed old Fig. 3d.

f. 4a: error bars are at null level?

There are no error bars for the large colored bars as they represent the p-value estimate from comparing two distributions as shown in the inset.

g. 4b (and following): no tick marks in the right plots

We corrected this error.

h. 5f: error bars

We opted to present the actual measurements as circles instead of error bars to convey the distribution of the measured data.

i. 6: add axis label, what type of data is shown?

Old Fig. 6 was replaced by new Fig. 6 and Fig. 7.

Reviewer #2 (Remarks to the Author):

In this manuscript from Vaknin et al, the authors develop a library of synthetic promoters by merging a target core promoter and upstream TF binding sites, to enhance protein expression in eukaryotic cells. This is not the first time synthetic promoters are designed by this approach and previous work has shown that is possible to engineer mammalian cells based on mining of TFBS and core promoter engineering (e.g. Johari et al. 2019 and others).

The manuscript tries to go beyond the state of art by building a system functional in more than one chassis. I have a few questions, mainly on the experimental side, as I am not a computational person. It is not fully clear to me what the authors mean with the term "motif" and it could be useful to define this.

We thank the reviewer for this very important comment and apologize for not specifying our rationale for choosing the motifs. In response to the comment, we have added a Methods subsection titled "Motif selection and encoding into OL", where our motif selection process and definition are described. In brief, motifs were based on short conserved sequence segments obtained by Jussi Taipale's group, and provided to us prior to their publication as position weight matrices (DOI: 10.1038/nbt.4138). The Taipale motifs were created by an empirical assay that was designed to test protein activity in a broad swath of organisms.

It is not fully clear also how the 41 motifs were identified and if this was validated by using more than one TF data base.

In brief, the Taipale study focused on developing a protein activity assay specifically for DNA-binding TFs in cell and tissue extracts. Based on the study's data, we selected 41 enriched motifs from various organisms and tissue cells (e.g., yeast, fly, and mice tissue cells), according to the following criteria:

3. Different motif types: 8 organism-shared motifs, 5 mice tissue-shared motifs, 14 unshared motifs (unique to an organism), and 14 unknown motifs (with unknown regulatory function).
4. Known/unknown regulatory function: 27 of the selected motifs have known regulatory function, and those were anticipated to be the control motifs. Characterizing the regulatory function of the 14 unknown motifs in yeast was one of the stated goals of the study.

With regards to validation by an additional database, this was not done for two major reasons. First, we chose the motifs based on the Taipale group experimental findings to show that our approach can be used as a validation exercise to similar experimental findings. Second, 14 of the motifs were uncharacterized and thus would not appear in any database. In fact, our assay enabled us to characterize the regulatory function of 5 of the 14 unknown motifs, and thus validated our approach.

In figure 1e it is not fully clear why authors have not taken population 2 and re-bin it in order to capture more variants and potentially more interesting candidates with higher expression. It could have been beneficial to do so here.

We fully agree with the reviewer's sentiment. Unfortunately, in this case, hindsight is 20/20. At the time of the actual SORT-seq experiment, we chose not to re-bin as we were worried about how the increased sorting time may affect the re-binned yeast cells. In addition, in preliminary tests we did not expect to find many variants in this upper bin, and as a result we opted not to risk adding expression noise to the re-binned variants at the expense of losing expression resolution. Since we were able to identify several strong up-regulating motifs, and train a machine-learning model which provided robust predictions for the unseen validation set, we are confident that whatever information was lost in our decision to not re-bin would not have affected our results or conclusions profoundly.

In figure 6, for CHO cell experiments, the authors mentioned they used a BFP for fluorescence normalisation. However, it is known from literature that resource competition can impact normalisation (see Frei et al, 2020. Jones et al, 2020).

We thank the reviewer for this comment and would like to clarify. The use of BFP in the CHO and HeLa cell experiments was done to ensure that cells that were identified as “red” (i.e. mCherry expressing) were indeed transfected by a plasmid and not false positives. In addition, for each variant we compared the mean mCherry expression levels measured to the ratio of the mCherry and BFP channels. The results (see the revised Supplementary Figure 6e) show that BFP does not impact the results or their interpretation.

Can the authors shown the expression levels of the CHO library but with no normalisation and check if that improves the results?

We thank the reviewer for this comment. In the revised manuscript, all mammalian data (see new Figs. 6 and 7) is presented without normalization by BFP.

Caption of figure 6 should also more clearly describe what the figures shows.

We thank the reviewer for this comment. The captions for revised Fig. 6 and 7 provide additional details about the data presented as compared with the captions in the original manuscript.

Minor comments pertain to several typos present in the manuscript like Eukaryotic that should be lower case; Figures are called at time with capital letter and at times with lower case.

We corrected the typos.

In conclusion I suggest the work to be published once these questions have been addressed.

Reviewers' Comments:

Reviewer #1:

Remarks to the Author:

The authors addressed all major and minor comments listed in the first round of reviews. The manuscript is greatly improved – I would recommend publication.

Reviewer #2:

Remarks to the Author:

First of all, I would like to thank the authors for addressing my comments and questions.

I am generally satisfied with their revision but I would need to ask clarification on two points.

Supplementary figure 6. The authors state that they are confirming that the expression of BFP does not impact mCherry expression. However, it is not very clear from the figure caption and response in the rebuttal, nor from the methods section, if they actually performed the experiments in presence and absence of competition or if they simply compared the mCherry signal alone with mCherry normalised on BFP but from the same experiment performed in presence of competition.

Can the authors clarify?

I would like also to ask the authors if they can comment on the large error bars present for HeLa cell expression experiments.

Reviewer #2

First of all, I would like to thank the authors for addressing my comments and questions.

I am generally satisfied with their revision but I would need to ask clarification on two points.

Reviewer comment: Supplementary figure 6. The authors state that they are confirming that the expression of BFP does not impact mCherry expression. However, it is not very clear from the figure caption and response in the rebuttal, nor from the methods section, if they actually performed the experiments in presence and absence of competition or if they simply compared the mCherry signal alone with mCherry normalised on BFP but from the same experiment performed in presence of competition. Can the authors clarify?

Author response: We thank the reviewer for the opportunity to clarify. We did not create a separate set of clones lacking the BFP gene. The plot presented in Supplementary Fig. 5e (note the renumbering of the supplementary figures) depicts the mCherry channel compared with the mCherry normalized by the BFP channel as measured on the same cells. The purpose of this plot is to show that normalizing by a synthetic house-keeping gene did not affect the trends observed in the data. We remind the reviewer that the utilization of this house-keeping gene was for us to ensure that sorted cells were transfected by the sURS plasmid. In our opinion an experiment lacking this form of “house-keeping” may not be as reliable, since “false-positive” cells may enter the analysis particularly for weakly expressing strains. We have added a statement regarding this experimental strategy to the Figure caption of supplementary figure 6 to alleviate any further confusion.

Reviewer comment: I would like also to ask the authors if they can comment on the large error bars present for HeLa cell expression experiments.

Author response: The error bar observed for the HeLa cells (i.e. Supplementary Fig. 5h) reflect the actual natural deviation in the expression data observed over the triplicates. The unusual variation may be due to the fact that the intensity of expression was not as strong as compared with the expression measured for the CHO cells, which may have led to increased noise. Fortunately, this expression level was sufficiently strong to differentiate between the different boosts enabled by the sURS variants, which correlated well with both the CHO cell measurements and the revised model.